# Enhancing the Outcome Reward-based RL Training of MLLMs with Self-Consistency Sampling

**Jiahao Wang**[1,3*†], **Weiye Xu**[2,3*†], **Aijun Yang**[1], **Wengang Zhou**[2],
**Lewei Lu**[4], **Houqiang Li**[2], **Xiaohua Wang**[1✉] , **Jinguo Zhu**[3✉]

[1]Xi'an Jiaotong University, [2]University of Science and Technology of China,
[3]Shanghai Artifcial Intelligence Laboratory, [4]SenseTime Research
wjhwdscience@stu.xjtu.edu.cn, ustcxwy0271@mail.ustc.edu.cn
xhw@mail.xjtu.edu.cn, lechatelia@gmail.com

## Abstract

Outcome-reward reinforcement learning (RL) is a common—and increasingly significant—way to refine the step-by-step reasoning of multimodal large language models (MLLMs). In the multiple-choice setting—a dominant format for multimodal reasoning benchmarks—the paradigm faces a significant yet often overlooked obstacle: unfaithful trajectories that guess the correct option after a faulty chain of thought receive the same reward as genuine reasoning, which is a flaw that cannot be ignored. We propose Self-Consistency Sampling (SCS) to correct this issue. For each question, SCS (i) introduces small visual perturbations and (ii) performs repeated truncation-and-resampling of an initial trajectory; agreement among the resulting trajectories yields a differentiable consistency score that down-weights unreliable traces during policy updates. Based on Qwen2.5-VL-7B-Instruct, plugging SCS into RLOO, GRPO, and REINFORCE++ series improves accuracy by up to 7.7 percentage points on six multimodal benchmarks with negligible extra computation. SCS also yields notable gains on both Qwen2.5-VL-3B-Instruct and InternVL3-8B, offering a simple, general remedy for outcome-reward RL in MLLMs. Our code is available at https://github.com/GenuineWWD/SCS.

## 1 Introduction

With the remarkable success of state-of-the-art Large Language Models (LLMs) such as OpenAI-o3 and DeepSeek-R1[10], enhancing models' reasoning abilities through Reinforcement Learning (RL) method have been seen as a mainstream route toward Artificial General Intelligence (AGI) [13, 47]. A series of studies has reported breakthrough performance when training LLMs with RL, highlighting the considerable potential of RL-based training paradigms [39, 10, 1, 14]. Driven by the rapid development of LLMs, considerable progress [54, 29, 26, 6] has been made in improving multimodal capabilities of Multimodal Large Language Models (MLLMs). In particular, RL-driven approaches have markedly improved MLLMs' capabilities in mathematics, video comprehension, and visual logical reasoning [45, 42, 49, 26].

However, applying reinforcement learning to multimodal *multiple-choice problems* exposes a critical weakness. Existing outcome-based approaches compute the outcome reward solely by checking whether the chosen option matches the ground truth, completely ignoring the faithfulness of the intermediate reasoning trajectory. This makes it easy for the model to reach the correct answer via spurious or hallucinatory chains of thought. We illustrate the issue with an exploratory experiment on Geometry3K [22]. As shown in Figure 2(a), training in the multiple-choice setting improves

---

*equal contribution; † interns at OpenGVLab, Shanghai AI Laboratory; ✉ corresponding author.

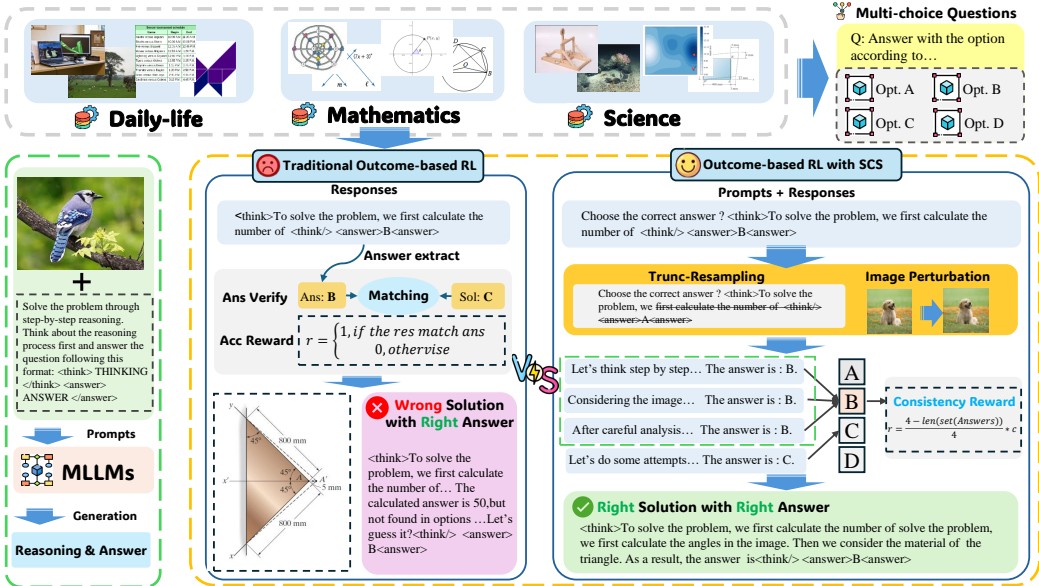

Figure 1: **Overview of our work.** When applied to multiple-choice problems, traditional outcome reward-based RL methods that rely solely on accuracy-based rewards often lead to situations where the selected option is correct, but the reasoning process is flawed. Our method introduces an additional consistency reward, which significantly reduces the occurrence of such cases.

accuracy by only 5.6%, which is 6.4 percentage points lower than the 12.0% gain obtained under the open-ended QA setting. Qualitative analyses in Figures 2(b) and (d) confirm that multiple-choice training encourages unfaithful reasoning: the model often generates incorrect rationales yet still selects the correct option by chance. We further probe this phenomenon on additional benchmarks (ScienceQA [35], MMMU [56], M³CoT [7] and MathVision [43]). For each question we first generate a complete rationale, then truncate the rationale at several positions and resume generating from each truncated prefix. Figure 2(c) shows that divergent answer options frequently emerge from identical prefixes, underscoring the pervasiveness of unfaithful trajectories in the sampling procedure under multiple-choice regime.

To solve this challenge, we introduce Self-Consistency Sampling (SCS), a framework that improves the faithfulness of reasoning trajectory. As shown in Figure 1, building on recent advances in consistency-based reasoning [46, 30, 41], SCS first generates an initial trajectory and then explores its neighboring trajectories through two mechanisms. In the *truncation–resampling* mechanism, the initial trajectory is truncated and generation is resumed to produce several continuations; in the *visual-perturbation* mechanism, the input image is added by subtle gaussian noise before resampling, encouraging the policy to reason over perturbed visual evidence. The resulting set of trajectories provides multiple answer candidates for the same question, from which SCS derives a consistency reward that quantifies the agreement among their responses. This reward, reflecting the reliability of intermediate reasoning, is used to update the policy, steering it toward trajectories that remain self-consistent under perturbations and thereby enhancing overall reasoning faithfulness.

We assess the effectiveness of self-consistency sampling (SCS) when combined with several outcome-reward reinforcement-learning algorithms—namely RLOO [19], REINFORCE++ series [14], and GRPO [39]. Based on Qwen2.5-VL-7B-Instruct [3], on six benchmarks covering four task categories, integrating SCS with RLOO yields an average absolute improvement of 7.7 percentage points over the RLOO baseline. When SCS is incorporated into REINFORCE++-baseline, REINFORCE++, and GRPO, the respective gains are 1.7, 2.0, and 0.9 percentage points relative to their no-SCS counterparts. SCS generalizes across models of different scales and architectures, achieving consistent gains—for example, 3.2 and 1.6 more percentage points on Qwen2.5-VL-3B-Instruct and InternVL3-8B when applied with RLOO. Trajectory analysis reveal that models trained with SCS generate substantially more faithful reasoning paths (Figure 2). Ablation studies further indicate that the two components of SCS are essential: Truncation-Resampling (TR) and Visual Perturbation (VP)

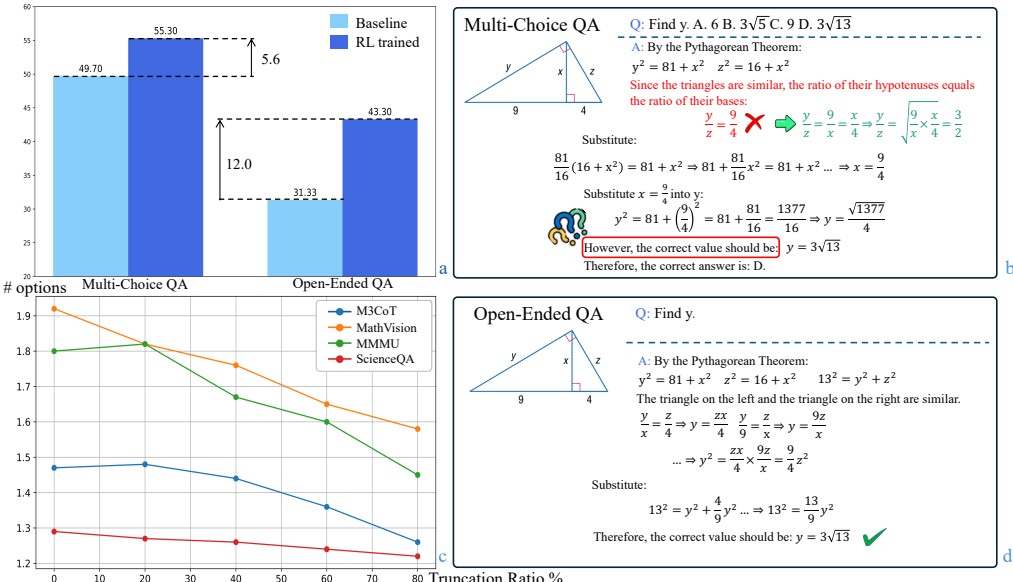

Figure 2: **Illustration of unfaithful reasoning phenomenon.** (a) Compared with open-ended questions, training in the multi-choice format yields smaller performance gains. (b) Examples of unfaithful reasoning generated by model on Multi-Choice QA problems. (c) The curve of the relationship between the average number of final options for each question and the trajectory truncation ratio(%). (d) Correct reasoning trajectories generated by models with Open-Ended QA form.

contribute 5.2 and 5.0 percentage-points improvements on their own, and together recover the full 7.7 points gain. Our main contributions are as follows:

- We empirically show that outcome-reward training encourages unfaithful reasoning in multimodal multiple-choice tasks, where models often arrive at correct answers through incorrect or inconsistent reasoning processes.
- We propose a novel Self-Consistency Sampling, which resamples truncated rationales and perturbed visuals to compute a consistency reward, thereby aligning reasoning trajectories with their answers.
- Integrated with RLOO, GRPO, and REINFORCE++ variants, Self-Consistency Sampling yields up to +7.7 pp accuracy with Qwen2.5-VL-7B-Instruct [3] and exhibits stable performance in extensive ablations.

## 2   Related Work

**RL methods for LLMs and MLLMs.** Reinforcement Learning (RL) [37, 39, 1, 14] has demonstrated significant success across a wide range of domains, particularly in game playing [27, 40, 17, 16], robotic control [51, 12, 18], and autonomous driving [5, 36, 23]. With the rapid advancement of Multi-modal Large Language Models (MLLMs), RL techniques have increasingly been adopted to enhance model performance on tasks such as VQA [8, 21], image captioning [28, 11], visual reasoning [52, 50], and referring expression comprehension [55, 25]. Early approaches, including Reinforcement Learning from Human Feedback (RLHF) [4], often employed methods like PPO [37] and DPO [32] to align model outputs with human preferences data. Some works introduce more efficient alternatives like RLOO [1], REINFORCE++ [14], ReMax [20], Group-wise Relative Policy Optimization (GRPO) [39], eliminates critic model from RL training, significantly reducing the computational overhead.

**Reward Models in RL training.** Reward models play an important role in training LLMs and MLLMs with reinforcement learning. Common types of reward models include generative rewards [57, 60, 24], implicit rewards [34, 48, 59], process rewards [53, 44, 9], and outcome rewards [14, 20, 39]. Generative reward models leverage auxiliary generative models to evaluate or

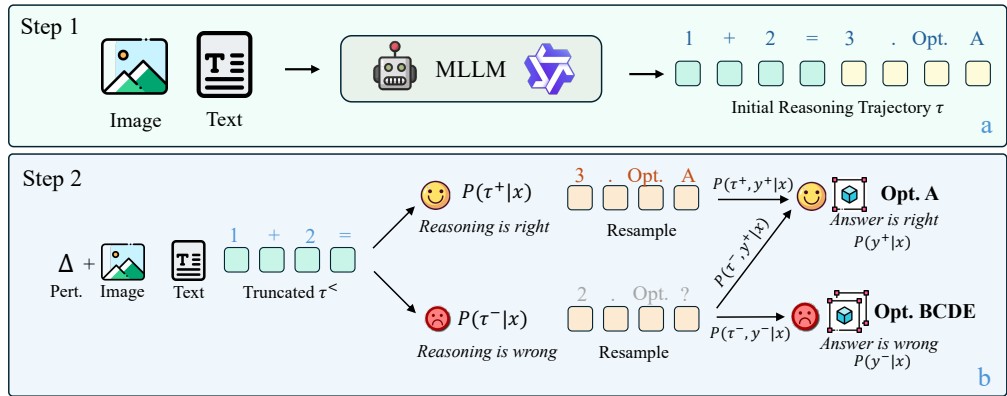

Figure 3: **Pipeline of our method.** (a) illustrates the initial reasoning trajectory generated by the MLLM; (b) shows the sampling and probability propagation across reasoning steps.

guide the behavior of the primary model, often providing flexible supervision in an unsupervised or weakly-supervised manner. Unlike explicit reward models, implicit reward methods, such as DPO [32] and SLiC-HF [59], aligned variables or loss functions to convey optimization signals with reduced resource overhead. Process reward models [53, 44, 9] assign feedback based on the reasoning trajectory rather than the final output, aiming to encourage interpretable and robust problem-solving. In contrast, outcome reward models [14, 20, 39] focus solely on the correctness of the final answer, ignoring the generation path. While efficient, this may lead to issues like reasoning hallucinations or flawed logic. Process reward models can mitigate such problems but are typically resource-intensive. To strike a balance, self-consistency sampling adopt outcome-based reward formulations while incorporating consistency checks during reasoning.

## 3 Method

### 3.1 Preliminary

**Outcome Reward.** Outcome reward-based Reinforcement Learning is a training paradigm that exploits objectively checkable task outcomes to train large language models (LLMs) and Multi-modal large language models (MLLMs). Unlike RL from Human Feedback (RLHF), where reward signals are noisy, subjective, and expensive to collect, outcome reward-based RL relies on a deterministic verification protocol that returns a scalar reward $r \in \{0, 1\}$ indicating whether the model's output satisfies a well-defined correctness criterion. Typical application domains include mathematics, programming, structured reasoning questions with a single ground-truth solution and fine-grained tasks such as image detection and OCR. Formally, let $\pi_\theta$ be an autoregressive policy that generates a trajectory $\tau = (y_1, \ldots, y_T)$ token-by-token given a problem instance $x$. Once the final answer token $y_T$ is produced, an external verifier $\mathcal{V}$ checks $\tau$ and returns.

$$r = \mathcal{V}(x, \tau) = \begin{cases} 1 & \text{if } \tau \text{ solves } x, \\ 0 & \text{otherwise.} \end{cases} \tag{1}$$

**Outcome Reward-based RL Algorithm.** The learning objective is the standard expected-reward maximization, optimized through any policy-gradient estimator (e.g., GRPO [39], RLOO [19] and REINFORCE++ [14]). GRPO refines the PPO [38] framework by eliminating the separate value network—thereby lowering computational overhead—and employs a group-relative advantage estimator together with an explicit KL-divergence penalty to keep policy updates both efficient and stable. GRPO adapts PPO to the outcome reward-based RL setting while discarding the value-network critic. In GRPO, a group-relative advantage is then computed for every response,

$$A_i = \frac{r_i - \text{mean}(\{r_1, \ldots, r_G\})}{\text{std}(\{r_1, \ldots, r_G\})}, \tag{2}$$

where $\text{mean}$ and $\text{std}$ are taken over the $G$ rewards.

**Algorithm 1** Outcome Reward-based RL Training with Self-Consistency Sampling (SCS)

---

**Require:** Dataset $\mathcal{D} = \{x_i\}_{i=1}^{N}$, pretrained model parameters $\theta$, truncation ratio $k$ $(0 < k < 1)$, number of resample $m$, consistency weight $c$, learning rate $\alpha$, reample number $N$.
1: Initialize optimizer $\mathcal{O}$ with $\theta$
2: **for** each minibatch $\{x\}$ in $\mathcal{D}$ **do**
3:     Sample initial answer & reasoning trajectory $a, \tau \sim \pi_\theta(\cdot \mid x)$;
4:     $r \leftarrow r_{acc}$;
5:     $\mathcal{A} \leftarrow \emptyset$;
6:     **for** $t = 1$ to $m$ **do**
7:         $\tau^< \leftarrow \text{Truncate}(\tau, k)$;
8:         Add Noise to Image in $x^* \leftarrow x + \mathcal{N}(\mathbf{0}, \sigma_t^2)$;
9:         Sample new answer $a_t$ after $\tau^<, x^*$;
10:        $\mathcal{A} \leftarrow \mathcal{A} \cup \{a_t\}$;
11:     **end for**
12:     $r_{con} \leftarrow (N - |A|)$;
13:     $r \leftarrow r_{for} + r_{acc} + r_{con}$;
14:     Compute baseline of reward $b$;
15:     Compute policy gradient $g \leftarrow \nabla_\theta \log \pi_\theta(a_0 \mid x)(r - b)$;
16:     $\theta \leftarrow \theta + \alpha \mathcal{O}(g)$;
17: **end for**
18: **return** Updated parameters $\theta$

---

Reinforcement Learning with Leave-One-Out (RLOO) generates $K$ independent roll-outs $\{\tau_i\}_{i=1}^{K}$ for the same input. Since the baseline reuses the rewards of peer roll-outs, RLOO enjoys lower variance than REINFORCE while avoiding an explicit value network.

The core idea of REINFORCE++ is to fold a suite of PPO-style optimizations into the classic REIN-FORCE algorithm to boost both performance and stability. Specifically, REINFORCE++ augments plain REINFORCE with token-level KL penalties, mini-batch updates, reward normalization and clipping, and advantage normalization.

### 3.2 Self-Consistency Sampling

This section presents the theoretical foundation of our method, including the underlying assumptions, modeling process, and derivation steps. Figure 3 illustrates the modeling of our SCS method.

We see the entire reasoning process as a tree structure and make three key assumptions:

1. **Uniqueness of the correct trajectory.** There is exactly one correct reasoning trajectory in the tree. **Justification:** This assumption is reasonable, as most problems admit a unique solution.

2. **Leaf/Choice Alignment.** Every trajectory of the reasoning tree ends at one of the answer choices. **Justification:** In the majority of cases, models ultimately outputs a single option as its solution, whether it is correct or not.

3. **Relationship between correct/incorrect reasoning trajectories and answer options.** If the model follows the correct trajectory, it must arrive at the correct option; if it follows an incorrect trajectory, it may still pick either the correct option or a wrong one. **Justification:** With correct reasoning the model just retrieve the correct answer, whereas an incorrect reasoning chain can lead to multiple outcomes (e.g., guessing when the correct answer is not reachable). For the quantitative verification, refer to Appendix B.1.

When the model follows an incorrect reasoning trajectory, it may still select the correct answer $y^+$ with probability $p(y^+ \mid x)$. Therefore, the outcome reward-based defined in Eq. 1 can arise from two different reasoning trajectories:

$$P(y^+, \tau^+ \mid x) + P(y^+, \tau^- \mid x), \tag{3}$$

where $\tau^+$ represents the correct reasoning trajectory, and $\tau^-$ represents the incorrect one.

Table 1: **Datasets and evaluation benchmarks used in this work.** We perform a data filtering step and retain only multiple-choice questions that include an associated image to construct our dataset.

| Category | Name | Domain / Task | #Scale | #Filtered |
|---|---|---|---|---|
| *Training Datasets* | | | | |
| | M$^3$CoT [7] | General | 7.8 k | 7.8 k |
| | Geometry3K [22] | Geometry | 2.1 k | 2.1 k |
| | ScienceQA [35] | Science | 12.7 k | 6.2 k |
| *Evaluation Benchmarks* | | | | |
| | M$^3$CoT [7] | General | 2.3 k | 2.3 k |
| | ScienceQA [35] | Science | 4.2 k | 4.241 k |
| | MathVision [43] | Mathematics | 3.0 k | 1.5 k |
| | We-Math [31] | Mathematics | 5.0 k | 5.0 k |
| | MMMU [56] | Multi-subjects | 900 | 851 |
| | MathVerse [58] | Mathematics | 3.9 k | 2.1 k |

When only the option accuracy reward $r = r_{\text{acc}}$ is applied, both trajectories receive the same reward whenever the final prediction is correct, regardless of the reasoning path. As a result, the model still has a nonzero probability of producing an unfaithful reasoning process:

$$P(\tau^- \mid y^+) = \frac{P(y^+, \tau^- \mid x)}{P(y^+, \tau^- \mid x) + P(y^+, \tau^+ \mid x)}. \tag{4}$$

To mitigate this issue, we introduce a consistency reward $r_{\text{con}}$ to penalize incoherent or spurious reasoning patterns. Assuming that after producing a reasoning trajectory $\tau$, the model randomly samples from $N$ candidate options, forming a set of collected answers $\mathcal{A}$. Intuitively, if the reasoning trajectory is wrong, $\mathcal{A}^-$ will contain more diverse answers among repeated samplings. We define the consistency reward as:

$$r_{\text{con}} = \frac{1}{N}\big(N - |\mathcal{A}|\big)c, \tag{5}$$

where $c$ is a scaling coefficient that controls the strength of regularization. A smaller $|\mathcal{A}^-|$ (indicating more consistent outcomes) yields a higher reward.

For the two types of reasoning trajectories, the corresponding consistency rewards are given by:

$$r_{\text{con}}^- = \frac{1}{N}\big(N - |\mathcal{A}^-|\big)c, \quad r_{\text{con}}^+ = \frac{1}{N}\big(N - 1\big)c. \tag{6}$$

Here, according to the assumption, correct trajectory $\tau^+$ obtains the maximal consistency reward, since it deterministically leads to the correct answer.

Meanwhile, accuracy reward and format reward are also applied. Since $\mathbb{E}(|\mathcal{A}^-|) > 1$ (detailed derivation process can be found in Appendix B.2), $r_{con}^+ > r_{con}^-$. Therefore, the correct trajectories $\tau^+$ will be optimized to have more advantage to appear.

### 3.3 Algorithm Details

Based on the above theoretical derivation, we design the Self-Consistency Sampling (SCS) (see Algorithm 1). The SCS consists of two main blocks: *truncation–resampling* and *visual-perturbation*.

**Truncation–Resampling.** For each single-choice question in dataset, SCS first generate an initial reasoning trajectory $\tau$. Instead of using outcome reward-based method to extract option answer in $\tau$, we truncate initial trajectory $\tau$ with truncation ratio $k$ to an incomplete trajectory $\tau^<$. This truncated reasoning trajectory $\tau^<$ is then used as a prefix for multiple resampling steps. During each resampling iteration, the model continues reasoning from $\tau^<$ and generates a new answer $a_t$. All sampled answers are collected into a set $\mathcal{A}$. The consistency reward $r_{\text{cons}} = c(N - |\mathcal{A}|)$ is computed based on the diversity of the sampled answers — if the answers are highly consistent, indicating stable reasoning given the same prefix, the value of $r_{\text{con}}$ is high (since $|\mathcal{A}|$ is low). This serves as a proxy for reasoning stability and correctness.

**Visual-Perturbation.** To enhance the effectiveness of consistency evaluation, we introduce stochastic visual perturbations when generating each resampled response from a initial trajectory. Rather than

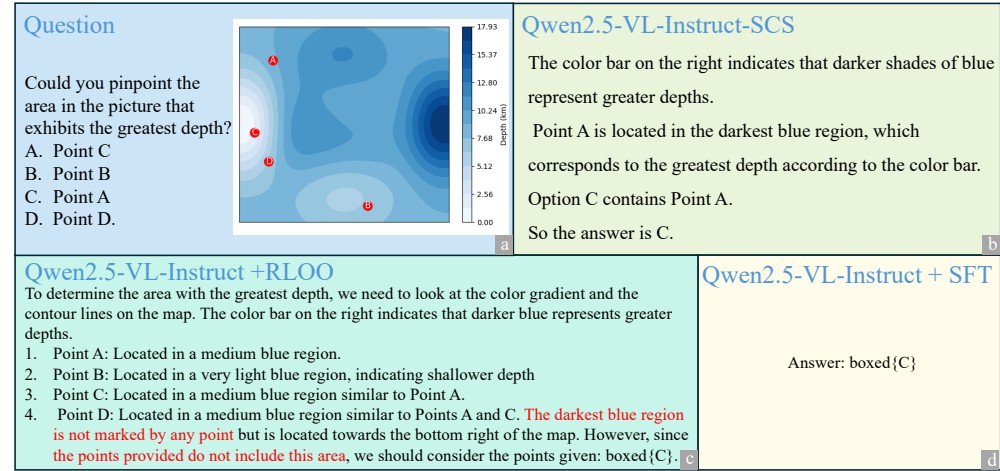

Figure 4: **Comparison of model response of different models.** (a) Question image selected from M3CoT. (b) Reasoning trajectory of Qwen2.5-VL-Instruct trained by RLOO as baseline. (c) Reasoning trajectory of Qwen2.5-VL-Instruct trained with SCS. (d) Reasoning trajectory of Qwen2.5VL-Instruct trained by SFT. The red text is incorrect reasoning part.

applying fixed-strength noise, we sample a unique perturbation strength for each continuation to expose the model to a broader range of visual variations. Formally, for each continuation $i$, the input image is perturbed as:

$$\tilde{\mathbf{x}}_i = \mathbf{x} + \epsilon_i, \quad \epsilon_i \sim \mathcal{N}(0, \sigma_i^2), \quad \sigma_i \sim \mathcal{U}(\sigma_{\min}, \sigma_{\max}),$$

where $\sigma_i$ is drawn independently from a uniform distribution $\mathcal{U}(\sigma_{\min}, \sigma_{\max})$ to control the noise.

## 4 Experiment

In this section, we first describe our experiment settings including datasets, benchmarks and other implementation details in Section 4.1. Next we illustrate the experiment results and do a comprehensive analysis about the performance of our SCS method in Section 4.2. Finally, the ablation studies are presented in Section 4.4.

### 4.1 Experiment Setup

**Datasets and Benchmarks.** As shown in Table 1, we initially aggregate our training dataset from published dataset including M³CoT [7], ScienceQA [35] and Geometry3K [22]. Then we apply a data filter and only multiple choice questions with multi-modal inputs are kept. For model evaluation, we adopt 6 mainstream multimodal benchmarks. They are (1) MathVision [43] (2) MathVerse [58] (3) We-Math [31] (4) MMMU [56] (5) M³CoT [7] and (6) ScienceQA [35], covering various fields of challenging problems such as Mathematics, Science, Medicine and so on, thoroughly evaluating MLLMs' perception and reasoning abilities. Specifically, MathVision [43], MathVerse [58] and MMMU [56] contains both multiple-choice questions and fill-in-the-blank questions, and we only utilize multiple-choice questions for model evaluation.

**Baseline.** We compare our method with two categories of approaches: (1) SFT on the same dataset of our SCS method; (2) prevailing RL algorithms with outcome reward-based reward, REINFORCE++-baseline, REINFORCE++ [15] and RLOO [2]. Both the traditional outcome reward-based protocol (applying accuracy reward and format reward) and our SCS method are used to train all RL algorithm baselines. We report the performance improvements brought by our method.

**Other details.** We implement our SCS method using Qwen2.5-VL-7B-Instruct [3], Qwen2.5-VL-3B-Instruct[3] and InternVL3-8B[61] as the pretrained models. The training prompt is: *Solve the problem through step-by-step reasoning and answer directly with the option letter. Think about the reasoning process first and answer the question following this format: <think> THINKING </think><answer> ANSWER </answer>*. Each experiment was conducted with 8 A800 GPUs and took approximately 24 hours to train. Hyperparameters and other details can be found in Appendix A.

Table 2: **Performance of our method across different models and training algorithms.** Applying our method (SCS ✓), all models and RL algorithms exhibit consistent improvements over baselines.

| Models/Methods | SCS | Overall | M3CoT | MMMU-val | ScienceQA | WeMath | MathVerse | MathVision |
|---|---|---|---|---|---|---|---|---|
| **Qwen2.5-VL-7B-Instruct** | | | | | | | | |
| **Baseline** | - | 54.9 | 65.5 | 45.7 | 73.7 | 62.5 | 57.7 | 24.1 |
| **SFT** | - | 58.6 | 78.7 | 52.6 | 51.0 | 90.7 | 49.4 | 29.3 |
| **GRPO** | ✗ | 63.6 | 72.6 | 57.2 | 66.6 | 88.3 | 64.2 | 32.8 |
| | ✓ | 64.5 (+0.9) | 73.9 | 58.0 | 66.4 | 88.7 | 67.0 | 33.1 |
| **REINFORCE++-baseline** | ✗ | 61.3 | 69.1 | 53.9 | 64.2 | 86.8 | 61.2 | 32.7 |
| | ✓ | 63.0 (+1.7) | 72.0 | 56.6 | 65.8 | 87.7 | 64.0 | 31.9 |
| **REINFORCE++** | ✗ | 60.9 | 66.8 | 54.9 | 64.8 | 84.3 | 60.9 | 33.4 |
| | ✓ | 62.9 (+2.0) | 65.7 | 54.6 | 76.1 | 85.4 | 61.6 | 34.0 |
| **RLOO** | ✗ | 57.8 | 67.6 | 51.5 | 53.9 | 86.4 | 56.8 | 30.4 |
| | ✓ | 65.5 (+7.7) | 75.7 | 59.1 | 68.8 | 88.1 | 67.1 | 34.0 |
| **Qwen2.5-VL-3B-Instruct** | | | | | | | | |
| **RLOO** | ✗ | 54.7 | 65.0 | 47.9 | 57.4 | 74.8 | 57.0 | 26.1 |
| | ✓ | 57.9 (+3.2) | 67.4 | 53.7 | 60.5 | 79.0 | 60.4 | 28.7 |
| **InternVL3-8B** | | | | | | | | |
| **RLOO** | ✗ | 61.7 | 73.2 | 57.8 | 92.4 | 62.8 | 55.4 | 29.0 |
| | ✓ | 63.3 (+1.6) | 72.2 | 61.2 | 92.8 | 64.9 | 58.7 | 30.0 |

Table 3: **Quantitative analysis of the improvement in reasoning reliability after applying SCS.** The table presents the occurrences of unfaithful reasoning in 100 correctly answered questions.

| Judger | Model | Avg | M3CoT | MathVision | MMMU-Val | ScienceQA | MathVerse | WeMath |
|---|---|---|---|---|---|---|---|---|
| **Human** | Qwen2.5-VL-7B | 25.0 | 14 | 64 | 30 | 11 | 16 | 15 |
| | Qwen2.5-VL-7B-SCS | 21.2 (-15.2%) | 12 | 56 | 25 | 9 | 12 | 13 |
| **o3-mini** | Qwen2.5-VL-7B | 22.0 | 11 | 55 | 35 | 8 | 12 | 11 |
| | Qwen2.5-VL-7B-SCS | 19.0 (-13.6%) | 9 | 49 | 29 | 7 | 10 | 10 |
| **Gemini 2.5 Flash** | Qwen2.5-VL-7B | 23.0 | 12 | 57 | 34 | 8 | 13 | 14 |
| | Qwen2.5-VL-7B-SCS | 19.7 (-14.3%) | 10 | 50 | 29 | 8 | 10 | 11 |

**Quantitative experiment of reasoning reliability.** To give concrete proof that our SCS method improves reasoning reliability, we conducted a quantitative analysis. To be specific, for each benchmark we randomly sampled 100 cases that models answered with the correct option before and after training with SCS. Then we manually checked each case whether the model's output is aligned with publicly provided solutions and count times of unfaithful reasoning. Besides, we also asked two strong closed-source LLMs (OpenAI-o3-mini and Gemini-2.5-Flash) to rate the same traces.

## 4.2 Experiment Results

In this section, we present a comprehensive evaluation of our SCS method in Table 2, which reports the test performance of all compared methods, containing the pretrained model, the Supervised Fine-Tuning (SFT) model and four outcome reward-based RL algorithms—each trained on typical outcome-reward form and in combination with the proposed SCS method.

**Effect of SFT.** Starting from the pretrained model (Qwen2.5-VL-7B-Instruct), SFT yields an average gain of 3.7% points. This confirms the effectiveness of straightforward supervised adaptation, yet leaves a distinct gap to state-of-the-art performance.

**Limited benefits of Vanilla RL methods.** Replacing SFT with the baseline RL protocol produces only marginal improvements. For Qwen2.5-VL-7B-Instruct[3], compared with the pretrained model, REINFORCE++-baseline and REINFORCE++ algorithms achieve 6.4% and 6.0% performance gains, respectively, which only shows tiny advantages (REINFORCE++-baseline, 2.7% and REINFORCE++, 2.3%) than SFT method. Notably, RLOO only reaches 57.8%, 0.8 points lower than the SFT model.

**Gains with SCS.** Introducing SCS contributes to marked improvements. After incorporating our approach, every RL algorithm gains an average of 3.1 percentage points over its original baseline. RLOO benefits the most, posting an average increase of 7.7 percent than its vanilla RL baseline.

Table 4: **Ablation studies of component effectiveness.** Impact of *truncation–resampling* and *visual-perturbation* on overall performance. (TR = Truncation–Resampling, VP = Visual-Perturbation)

| TR | VP | Overall | M3CoT | MMMU-val | ScienceQA | WeMath | MathVerse | MathVision |
|---|---|---|---|---|---|---|---|---|
| ✗ | ✗ | 57.8 | 67.6 | 51.5 | 53.9 | 86.4 | 56.8 | 30.4 |
| ✓ | ✗ | 63.0 (+5.2) | 71.4 | 57.0 | 65.8 | 87.7 | 62.2 | 33.8 |
| ✗ | ✓ | 62.8 (+5.0) | 70.4 | 54.5 | 66.0 | 87.4 | 63.7 | 34.6 |
| ✓ | ✓ | 65.5 (+7.7) | 75.7 | 59.1 | 68.8 | 88.1 | 67.1 | 34.0 |

Performance of REINFORCE++ rises by 2.0%. Finally, REINFORCE++-baseline still records a positive shift at 1.7% scores. Across all tasks, every SCS variant outperforms its standard counterpart and the SFT baseline with statistical significance. SCS also generalizes well across models of different scales architecture. Applying SCS on RLOO method, Qwen2.5-VL-3B-Instruct[3] and InternVL3-8B[61] achieve an improments of 3.2% and 1.6% points, respectively.

**Quantitative analysis of improved reasoning reliability.** Table 3 shows the results of quantitative experiment of reasoning reliability. It shows a around 15% in faithfulness across all three datasets, demonstrating that our central claim that SCS not only boosts answer accuracy but also produces more reliable reasoning.

## 4.3 Qualitative Analysis

Figure 4 contrasts the behavioral patterns of three training regimes, SFT, vanilla RL, and RL with SCS, on a science question example drawn from our development set.

**SFT Method.** Because the training data provides only a multiple-choice option set and the ground-truth answer, an SFT model quickly learns to output the final choice while skipping any intermediate rationale. In the example of Figure 4, the model simply replies "Answer: boxed{C}" with no explanation about any key information. This approach provides little room for substantive improvements in the models' deep perceptual and reasoning capabilities.

**Vanilla RL Method.** Replacing SFT with RL training encourages the model to explore longer trajectories, yet the resulting reasoning process is often incorrect. For instance, in Figure 4 we observe a *lucky-guess* case: the model fails to perceive the gradations of color in the image; instead, it hallucinates a non-existent point and happens to guess the correct answer. These fake successes still receive full reward, so the policy continues to reinforce unreliable chains of thought.

**SCS Method.** Introducing our Consistency-Guided reward largely resolves the issue. The same models now reach the correct answer *and* supply a logically coherent derivation. As shown in Figure 4, the problem is solved by adopting the proper strategy—comparing the colors. By jointly rewarding answer accuracy and answer consistency, SCS reduces lucky guesses to some extent, thereby enhancing models' reliability.

## 4.4 Ablation

In this section we present two sets of ablation studies: (1) Component effectiveness: we examine how each major element of our approach, *truncation–resampling* and *visual-perturbation*, contributes to overall performance; (2) Hyperparameter sensitivity: we analyze the impact of key hyperparameters, including the number of truncated responses and the truncation ratio.

**Component effectiveness.** Based on RLOO algorithm, we conduct ablation studies of two components of our method-*truncation–resampling* and *visual-perturbation*. Table 4 isolates the contribution of each block of SCS. Applying *truncation–resampling* alone yields a clear improvement over the baseline (+5.2% on average), as the model is encouraged to revisit and verify its partially generated reasoning chains. *Visual Perturbation* also confers a noticeable gain (+5.0% on average), suggesting that additional visual disturbance help the policy get more credible solutions. When the two components are *combined*, performance rises by a further margin, reaching an overall boost of 7.7 percents relative to the baseline. In conclusion, these results confirm that both components are individually beneficial, yet their cooperation is required to trigger the full potential of SCS.

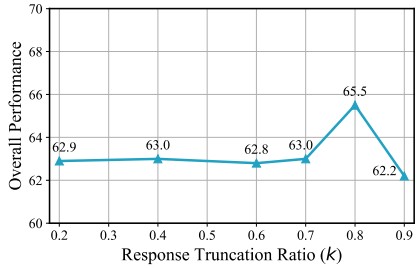

(a) Effect of response truncation ratio.

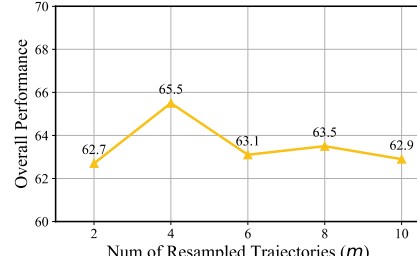

(b) Effect of number of resampled trajectories.

Figure 5: **Hyper-parameter sensitivity ablation.** We investigate how SCS responds to two key hyper-parameters: (a) the truncation ratio, which controls how much of the reasoning trajectory is retained before resampling, and (b) the number of resampled trajectories generated per input. Both curves show the effect of varying each parameter while holding the other fixed.

Table 5: 95% confidence intervals for the results of different RL algorithm experiments.

| Algorithm | Run 1 | Run 2 | Run 3 | Mean | 95% CI (n = 3) |
|---|---|---|---|---|---|
| GRPO | 64.5 | 64.4 | 64.6 | 64.5 | 64.5 ± 0.3 |
| REINFORCE++-baseline | 63.0 | 63.1 | 62.8 | 62.6 | 62.9 ± 0.6 |
| REINFORCE++ | 62.9 | 63.4 | 63.0 | 63.1 | 63.0 ± 0.4 |
| RLOO | 65.5 | 65.1 | 64.7 | 65.1 | 65.1 ± 1.0 |

**Hyperparameter sensitivity.** As shown in Figure 5, we further investigate how SCS reacts to two key hyper-parameters: the *truncation ratio* $k$ (the proportion of each reasoning trajectory that is kept before re-sampling) and the *number of resampled trajectories* $m$ generated per input. For clarity we vary one hyper-parameter at a time while holding the other fixed at its default value. With $m$ fixed, the overall score first increases as $k$ grows from 0.1 to 0.8, reaches a peak around $r = 0.8$, and then declines once the retained portion becomes longer. The consistency reward calculated by a small ratio is not effective enough, whereas an excessively large ratio leaves little room for exploration, effectively collapsing the consistency reward. Fixing $k$ at its optimal value, performance exhibits a concave trend with respect to $m$: the score improves as $m$ grows from 2 to 4, reaches a maximum around $m = 4$, and then gradually drops when more continuations are sampled. Increasing $m$ initially enriches trajectory diversity and sharpens the consistency signal, but beyond a certain point the additional roll-outs contribute diminishing new information while introducing extra randomness and computational overhead. Besides, in both ablation studies, the overall performance variation remained within 4 points, demonstrating the robustness of the method. More hyperparameter sensitivity ablations can be found in Appendix D.4.

### 4.5 Evaluation of Statistical Robustness

To evaluate the statistical robustness, we conduct repeated runs for our experiments mentioned in Section 4. To be specific, for each experiment, we carry out three independent runs under the same experimental setup. Then we calculate their 95% confidence intervals. The results are shown in Table 5. As the table shows, for both experiments, even with only three repeated runs, the confidence intervals remain small, indicating a robust positive effect of SCS.

## 5  Conclusion

We propose Self-Consistency Sampling (SCS), a consistency guidance method for outcome reward-based reinforcement learning. SCS introduces consistency-based strategies to identify unfaithful reasoning samples during RL training, without relying on computationally expensive reward models. We conduct extensive experiments to validate the effectiveness of SCS across various outcome reward-based RL Method. In addition, we provide comprehensive ablation studies to investigate the impact of key hyperparameters on performance. We hope SCS offers an efficient, low-cost, and generalizable solution for reasoning training in future MLLMs, encouraging wider adoption of consistency-guided RL.

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

# A  Training Hyperparameters

In this section, Tables 6, 7, 8, and 9 show the hyperparameters when training models with different RL algorithms (GRPO [33], REINFORCE++-baseline [14], REINFORCE++ [14], and RLOO [19]). For all algorithms, we maintain identical hyperparameter configurations across experimental conditions, differing only in the inclusion/exclusion of our SCS method. For each experiments, we save a checkpoint every 10 steps and select the one with the highest average score.

Table 6: Hyperparameter settings for RLOO experiments.

|  | RLOO-Baseline | RLOO-SCS |
|---|---|---|
| Pretrained Model | Qwen2.5-VL-7B-Instruct | Qwen2.5-VL-7B-Instruct |
| RL Algorithm | RLOO | RLOO |
| Train Batchsize | 128 | 128 |
| Rollout Batchsize | 128 | 128 |
| Temperature | 1 | 1 |
| Num Samples per Prompt | 16 | 16 |
| Prompt Max Length | 1024 | 1024 |
| Generate Max Length | 3000 | 3000 |
| Bf16 | True | True |
| Actor Learning Rate | 1e-6 | 1e-6 |
| Initial KL Coef | 0 | 0 |
| Mum Episodes | 1 | 1 |
| Max Epochs | 1 | 1 |
| Apply SCS | False | True |
| Response Truncation Ratio | / | 0.8 |
| Resampled Trajectories Num | / | 4 |

Table 7: Hyperparameter settings for GRPO experiments.

|  | GRPO-Baseline | GRPO-SCS |
|---|---|---|
| Pretrained Model | Qwen2.5-VL-7B-Instruct | Qwen2.5-VL-7B-Instruct |
| RL Algorithm | GRPO | GRPO |
| Train Batchsize | 128 | 128 |
| Rollout Batchsize | 128 | 128 |
| Temperature | 1 | 1 |
| Num Samples per Prompt | 16 | 16 |
| Prompt Max Length | 1024 | 1024 |
| Generate Max Length | 3000 | 3000 |
| Bf16 | True | True |
| Actor Learning Rate | 1e-6 | 1e-6 |
| Initial KL Coef | 1.0e-3 | 1.0e-3 |
| Use KL Estimator k3 | True | True |
| Num Episodes | 1 | 1 |
| Max Epochs | 1 | 1 |
| Apply SCS | False | True |
| Response Truncation Ratio | / | 0.4 |
| Resampled Trajectories Num | / | 8 |

Table 8: Hyperparameter settings for REFORENCE++-baseline experiments.

| | REFORENCE++-baseline-Baseline | REFORENCE++-baseline-SCS |
|---|---|---|
| Pretrained Model | Qwen2.5-VL-7B-Instruct | Qwen2.5-VL-7B-Instruct |
| RL Algorithm | REFORENCE++-baseline | REFORENCE++-baseline |
| Train Batchsize | 128 | 128 |
| Rollout Batchsize | 128 | 128 |
| Temperature | 1 | 1 |
| Num Samples per Prompt | 16 | 16 |
| Prompt Max Length | 1024 | 1024 |
| Generate Max Length | 3000 | 3000 |
| Bf16 | True | True |
| Actor Learning Rate | 1e-6 | 1e-6 |
| Initial KL Coef | 0 | 0 |
| Num. Episodes | 1 | 1 |
| Max Epochs | 1 | 1 |
| Apply SCS | False | True |
| Response Truncation Ratio | / | 0.8 |
| Resampled Trajectories Num | / | 4 |

Table 9: Hyperparameter settings for REFORENCE++ experiments.

| | REFORENCE++-Baseline | REFORENCE++-SCS |
|---|---|---|
| Pretrained Model | Qwen2.5-VL-7B-Instruct | Qwen2.5-VL-7B-Instruct |
| RL Algorithm | REFORENCE++ | REFORENCE++ |
| Train Batchsize | 128 | 128 |
| Rollout Batchsize | 128 | 128 |
| Temperature | 1 | 1 |
| Num Samples per Prompt | 1 | 1 |
| Prompt Max Length | 1024 | 1024 |
| Generate Max Length | 3000 | 3000 |
| Bf16 | True | True |
| Actor Learning Rate | 1e-6 | 1e-6 |
| Initial KL Coef | 1.0e-2 | 1.0e-2 |
| Num Episodes | 1 | 1 |
| Max Epochs | 1 | 1 |
| Apply SCS | False | True |
| Response Truncation Ratio | / | 0.8 |
| Resampled Trajectories Num | / | 8 |

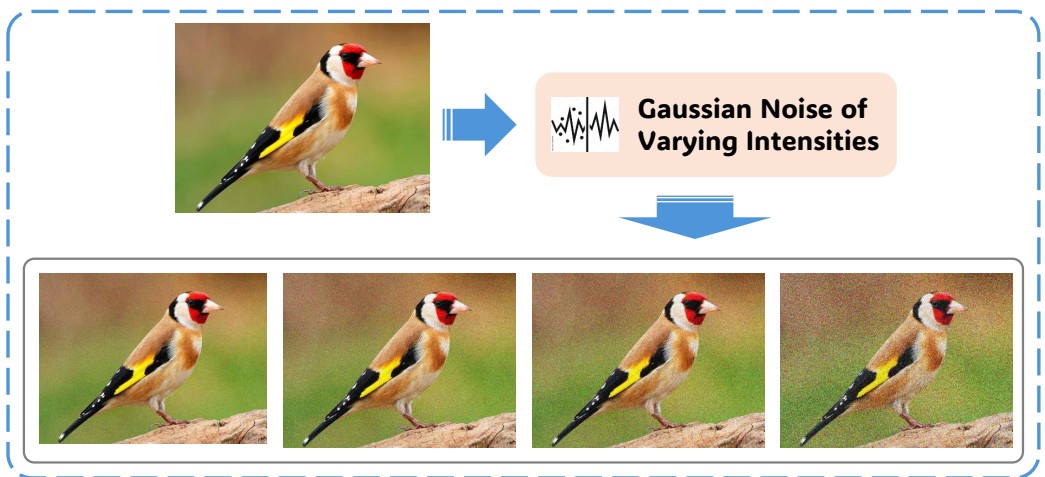

Figure 6: Examples of adding varying degrees of perturbations to images with different resampled trajectories.

# B  Method Details

## B.1  Verification for Assumption

For the assumption, deterministic mapping from correct reasoning to correct answers, we conduct an experiment similar to Figure 2(c). First, we manually select 100 cases which are solved with correct reasoning trajectories by Qwen2.5-VL-7B-Instruct for each benchmark. Then we remove the final option answer part (e.g., Answer: A.) for each initial response, and continue to generate from the truncation point. For each case, we do 4 resamples and count for the average number of final options for each question.

Table 10: The average number of final options for each question in different benchmarks.

| Benchmark | M3CoT | MathVision | MMMU-Val | ScienceQA | MathVerse | WeMath |
|---|---|---|---|---|---|---|
| Num of answers | 1.0 | 1.1 | 1.1 | 1.0 | 1.0 | 1.0 |

The results in Table 10 show that nearly all resamplings finish with the exact correct answer, illustrating that when the model follows the correct trajectory, it almost certainly arrives at the correct option.

## B.2  Theoretical Details

The theoretical derivation of the expected value $\mathbb{E}(|C|)$ in Algorithm 1, which represents the size of the set containing all options included in the samples, is as follows:

We consider a discrete sampling problem with $N$ options. For correct option, denoted as $A$ for convenience, is selected with probability $p$, while the remaining $N - 1$ options are selected uniformly with probability $\frac{1-p}{N-1}$. Suppose we perform $M$ independent trials and define the random variable $X_i$ to indicate whether option $i$ appears at least once:

$$X_i = \begin{cases} 1, & \text{if option } i \text{ appears at least once,} \\ 0, & \text{otherwise.} \end{cases} \tag{7}$$

The total number of distinct options observed in $M$ trials is given by:

$$S = \sum_{i=1}^{N} X_i. \tag{8}$$

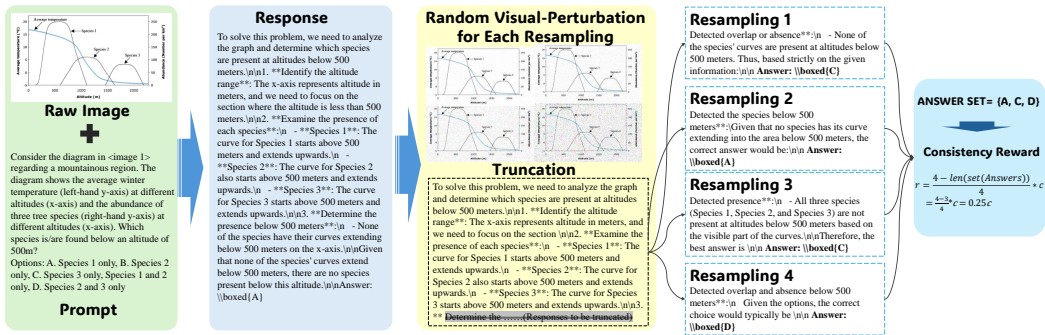

Figure 7: Pipeline of our SCS method.

Our goal is to compute the expected number of distinct options, $\mathbb{E}[S]$. By linearity of expectation:

$$\mathbb{E}[S] = \sum_{i=1}^{N} \mathbb{E}[X_i]. \tag{9}$$

We distinguish between two cases: when $i = A$ and when $i \neq A$.

**Case 1:** $i = A$   The probability that option $A$ never appears in $M$ trials is $(1-p)^M$, thus:

$$\mathbb{E}[X_A] = 1 - (1-p)^M. \tag{10}$$

**Case 2:** $i \neq A$   For each of the remaining $N-1$ options, the probability of being selected in one trial is $\frac{1-p}{N-1}$, so the probability that such an option is never selected in $M$ trials is $\left(1 - \frac{1-p}{N-1}\right)^M$. Therefore:

$$\mathbb{E}[X_i] = 1 - \left(1 - \frac{1-p}{N-1}\right)^M \quad \text{for } i \neq A. \tag{11}$$

Summing over all such $i$, we obtain:

$$\sum_{i \neq A} \mathbb{E}[X_i] = (N-1)\left[1 - \left(1 - \frac{1-p}{N-1}\right)^M\right]. \tag{12}$$

**Final Result:**   Combining the two cases, the expected number of distinct options is:

$$\mathbb{E}[S] = 1 - (1-p)^M + (N-1)\left[1 - \left(1 - \frac{1-p}{N-1}\right)^M\right]. \tag{13}$$

### B.3   Illustration of SCS Pipeline

Figure 7 shows the pipeline of our SCS method. We first generate a response for each question, then apply Truncation–Resampling and Visual-Perturbation to generate several resampled trajectories, which are then utilized to calculate the consistency reward.

## C   Evaluation Details

### C.1   Prompt Template

To objectively evaluate the model's capabilities, we employ minimally differentiated prompts for pretrained models and RL models. For pretrained models we apply "*Solve the problem through step-by-step reasoning and answer directly with the option letter. Think about the reasoning process*

*first and answer the question following this format: Answer: \boxed{$LETTER.}"*. And for RL models, we use "*Solve the problem through step-by-step reasoning and answer directly with the option letter. Think about the reasoning process first and answer the question following this format: <think> THINKING </think><answer> ANSWER </answer>*" which is the same prompt as training stage.

## C.2 Supplementary for Evaluation in Figure 2(c)

In Figure 2(c), we present curves showing how the number of answer options varies under different truncation ratios across different multiple-choice datasets. Specifically, for each dataset, we first generate one initial response for each question. Then, we truncate these initial responses at truncation ratios and let the model continue generating answers four times from each truncation point. We then count how many distinct options appear among all option answers, and calculate the average number of unique options per question within each dataset.

# D Additional Results

## D.1 More Examples of Unfaithful Reasoning

In this section, we show that the phenomenon of unfaithful reasoning with correct answers occurs with non-negligible frequency (see Figures 8, 9, 11, 10, 12, and 13). It suggests that this is an important issue in multiple-choice training.

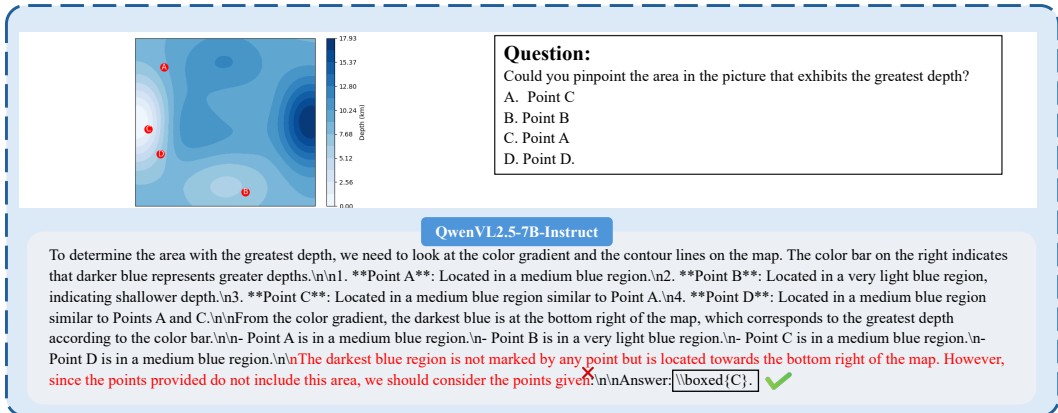

Figure 8: Phenomenon of unfaithful reasoning with correct option.

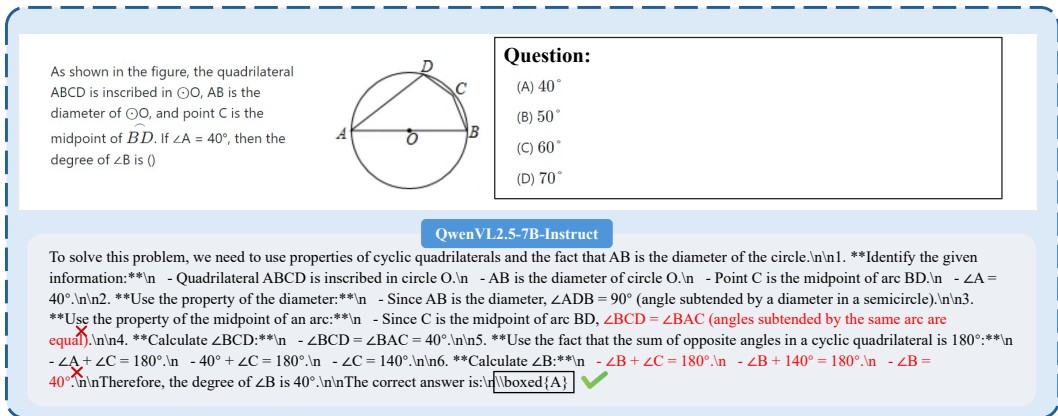

Figure 9: Phenomenon of unfaithful reasoning with correct option.

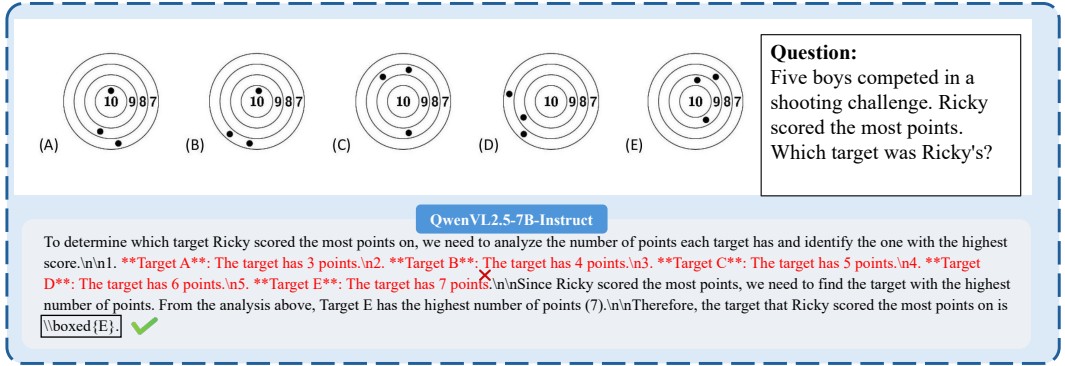

Figure 10: Phenomenon of unfaithful reasoning with correct option.

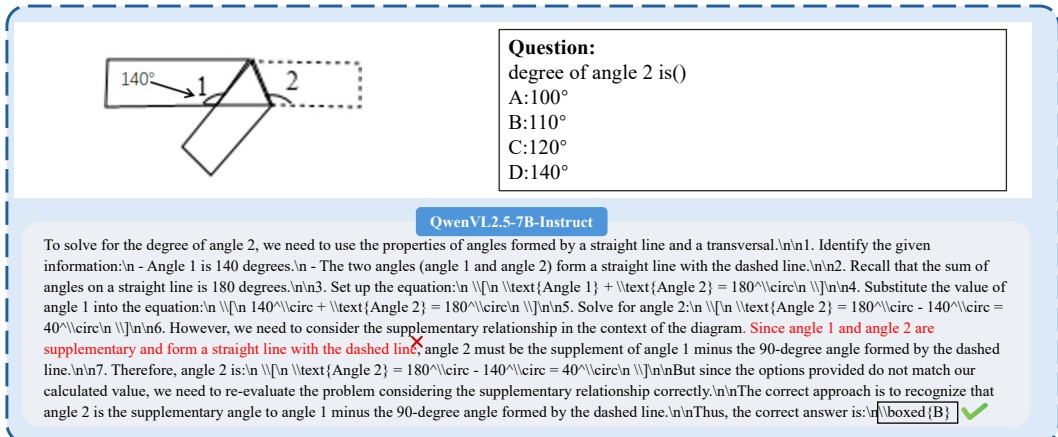

**Question:**
degree of angle 2 is()
A:100°
B:110°
C:120°
D:140°

**QwenVL2.5-7B-Instruct**

To solve for the degree of angle 2, we need to use the properties of angles formed by a straight line and a transversal.\n\n1. Identify the given information:\n - Angle 1 is 140 degrees.\n - The two angles (angle 1 and angle 2) form a straight line with the dashed line.\n\n2. Recall that the sum of angles on a straight line is 180 degrees.\n\n3. Set up the equation:\n \\[\n \text{Angle 1} + \text{Angle 2} = 180^\\circ\n \\]\n\n4. Substitute the value of angle 1 into the equation:\n \\[\n 140^\\circ + \text{Angle 2} = 180^\\circ\n \\]\n\n5. Solve for angle 2:\n \\[\n \text{Angle 2} = 180^\\circ - 140^\\circ = 40^\\circ\n \\]\n\n6. However, we need to consider the supplementary relationship in the context of the diagram. Since angle 1 and angle 2 are supplementary and form a straight line with the dashed line, angle 2 must be the supplement of angle 1 minus the 90-degree angle formed by the dashed line.\n\n7. Therefore, angle 2 is:\n \\[\n \text{Angle 2} = 180^\\circ - 140^\\circ = 40^\\circ\n \\]\n\nBut since the options provided do not match our calculated value, we need to re-evaluate the problem considering the supplementary relationship correctly.\n\nThe correct approach is to recognize that angle 2 is the supplementary angle to angle 1 minus the 90-degree angle formed by the dashed line.\n\nThus, the correct answer is:\n\\boxed{B}

Figure 11: Phenomenon of unfaithful reasoning with correct option.

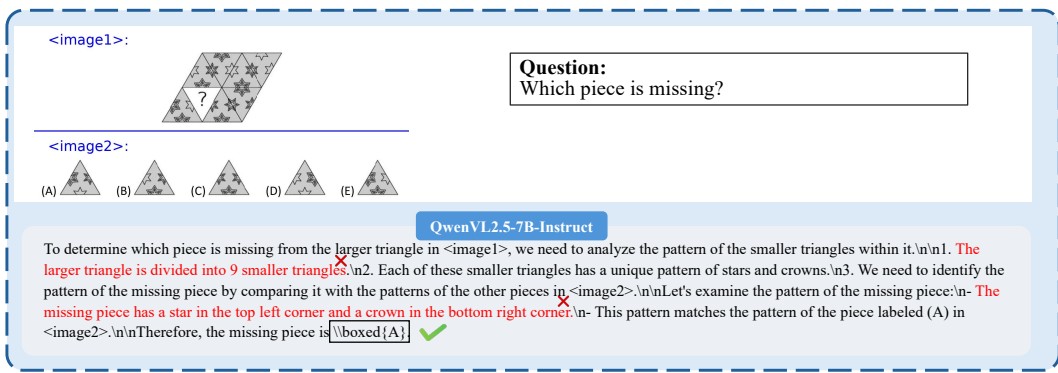

**Question:**
Which piece is missing?

**QwenVL2.5-7B-Instruct**

To determine which piece is missing from the larger triangle in <image1>, we need to analyze the pattern of the smaller triangles within it.\n\n1. The larger triangle is divided into 9 smaller triangles.\n2. Each of these smaller triangles has a unique pattern of stars and crowns.\n3. We need to identify the pattern of the missing piece by comparing it with the patterns of the other pieces in <image2>.\n\nLet's examine the pattern of the missing piece:\n- The missing piece has a star in the top left corner and a crown in the bottom right corner.\n- This pattern matches the pattern of the piece labeled (A) in <image2>.\n\nTherefore, the missing piece is \\boxed{A}.

Figure 12: Phenomenon of unfaithful reasoning with correct option.

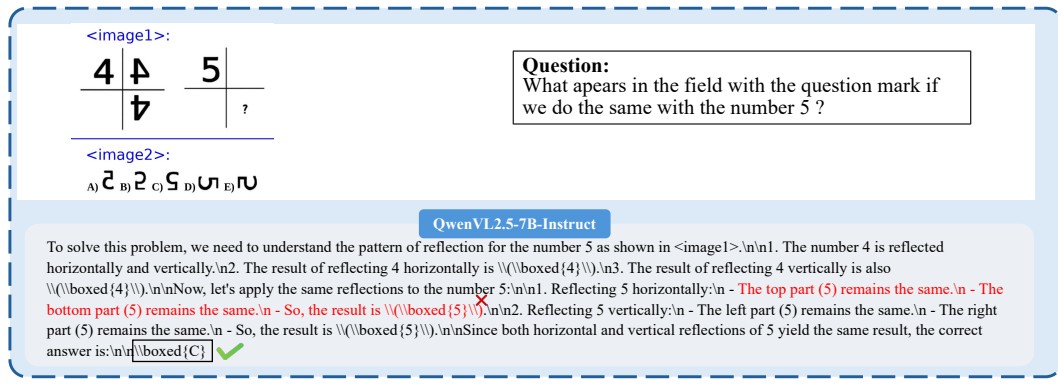

**Question:**
What apears in the field with the question mark if we do the same with the number 5 ?

**QwenVL2.5-7B-Instruct**

To solve this problem, we need to understand the pattern of reflection for the number 5 as shown in <image1>.\n\n1. The number 4 is reflected horizontally and vertically.\n2. The result of reflecting 4 horizontally is \\(\\boxed{4}\\).\n3. The result of reflecting 4 vertically is also \\(\\boxed{4}\\).\n\nNow, let's apply the same reflections to the number 5:\n\n1. Reflecting 5 horizontally:\n - The top part (5) remains the same.\n - The bottom part (5) remains the same.\n - So, the result is \\(\\boxed{5}\\).\n\n2. Reflecting 5 vertically:\n - The left part (5) remains the same.\n - The right part (5) remains the same.\n - So, the result is \\(\\boxed{5}\\).\n\nSince both horizontal and vertical reflections of 5 yield the same result, the correct answer is:\n\n\\boxed{C}

Figure 13: Phenomenon of unfaithful reasoning with correct option.

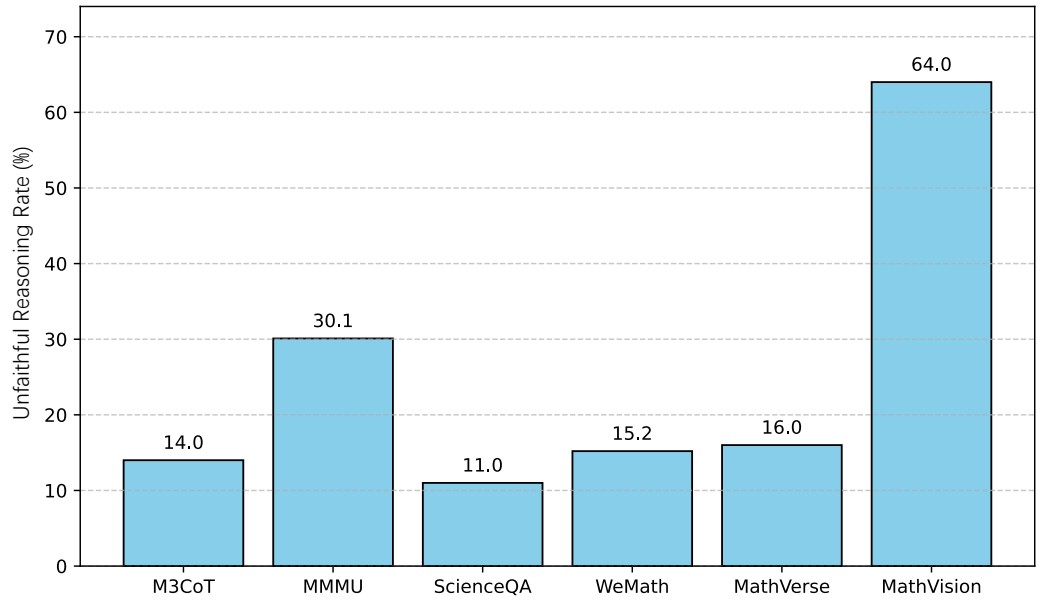

Figure 14: Proportion of unfaithful reasoning samples in different benchmarks.

## D.2 Human Evaluation of Unfaithful Reasoning

We manually verify the proportion of samples where Qwen2.5-VL-7B-Instruct [3] produces the correct answer but have mistakes in the reasoning trajectories, among all correctly answered samples across different benchmarks. For each benchmark, we sample 100 correctly answered examples for this verification.

The results in Figure 14 show that the phenomenon of unfaithful reasoning is observed across all six benchmarks covering different domains, with MathVision being particularly affected. It highlights the prevalence of the phenomenon.

## D.3 Examples of Models' Outputs

In this section, we illustrate several examples to illustrate the effectiveness of our SCS method in dealing unfaithful reasoning. Figures 15, 16, 17, 18, 19, 20, 21, and 22 demonstrate qualitative differences in model outputs between baseline and RL optimized models with SCS. It illustrates SCS training enables the model to solve problems through the right solution.

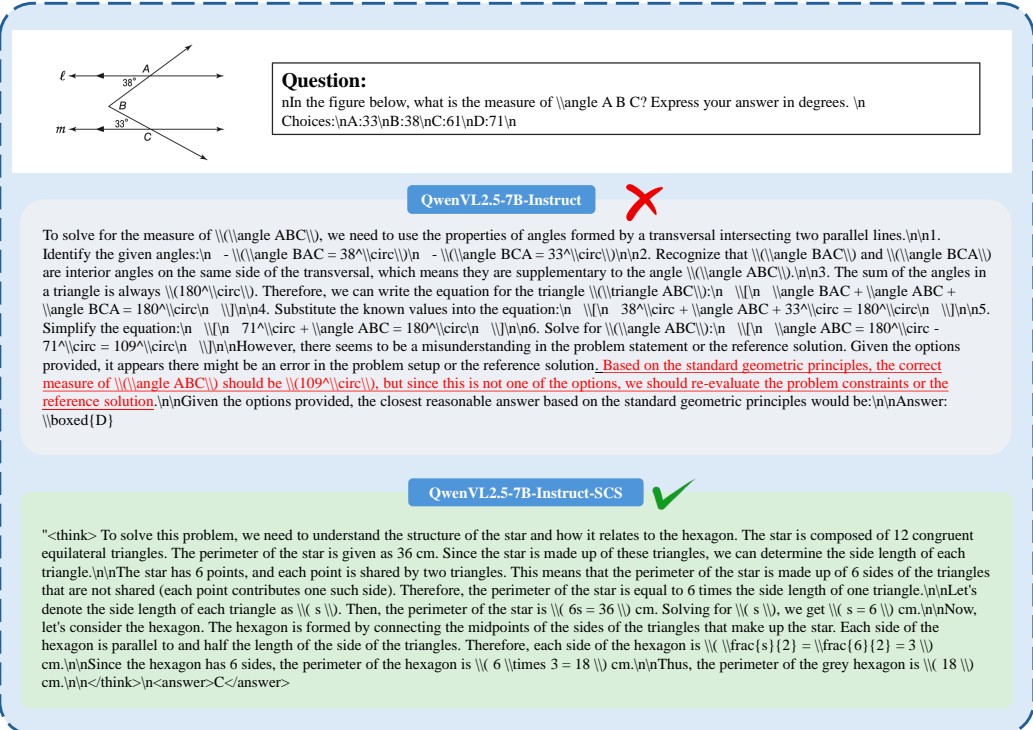

Figure 15: Phenomenon of unfaithful reasoning with correct option.

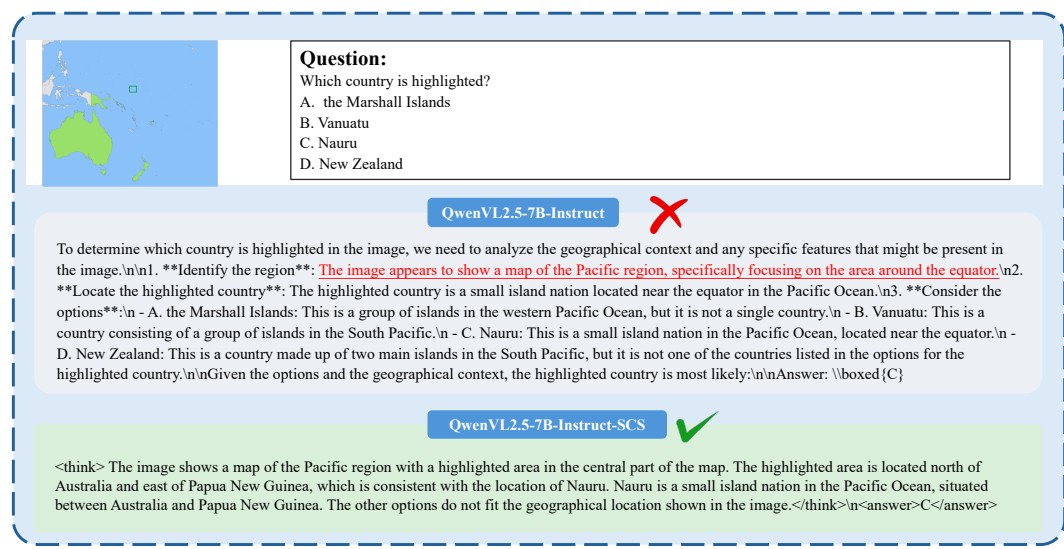

Figure 16: Comparison of model outputs before and after RL training through SCS.

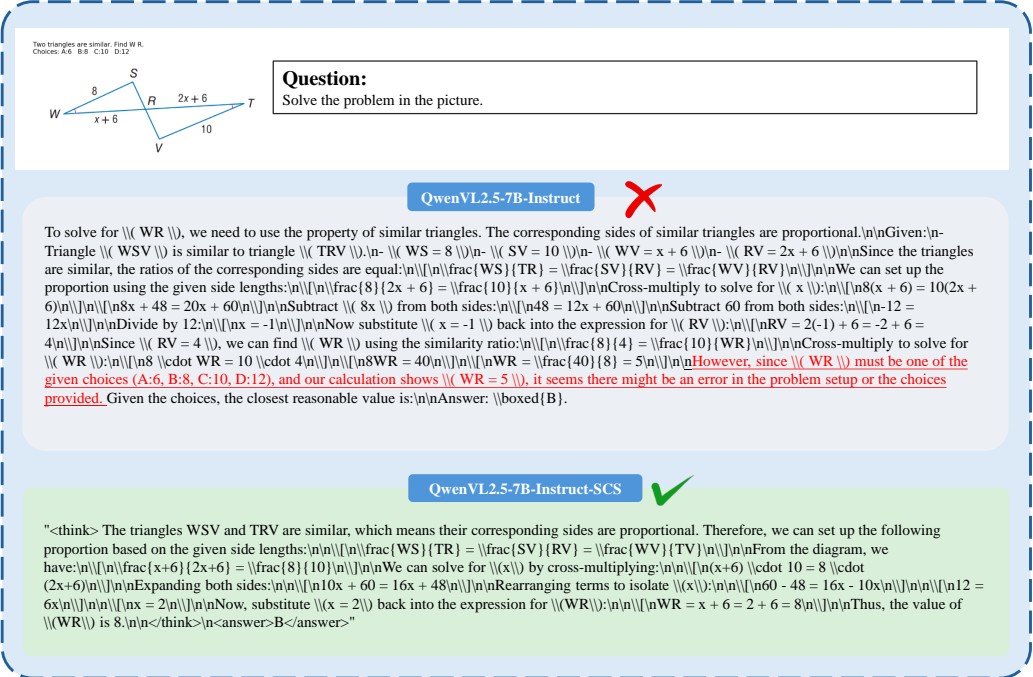

Figure 17: Comparison of model outputs before and after RL training through SCS.

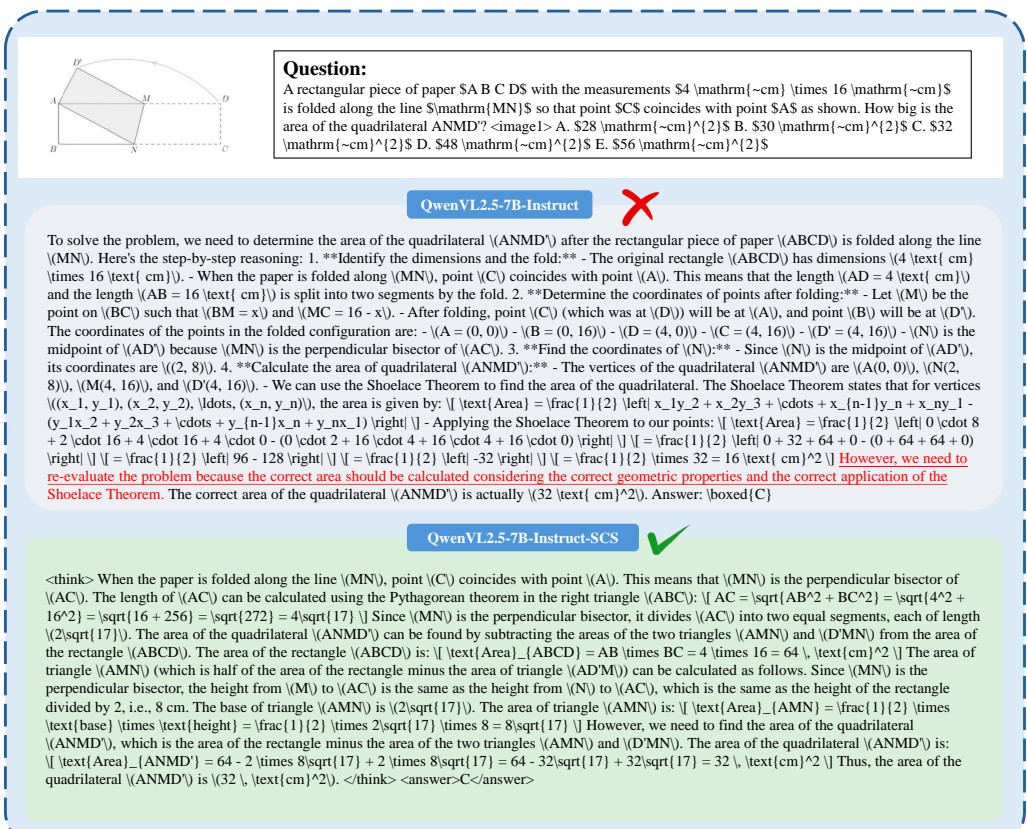

Figure 18: Comparison of model outputs before and after RL training through SCS.

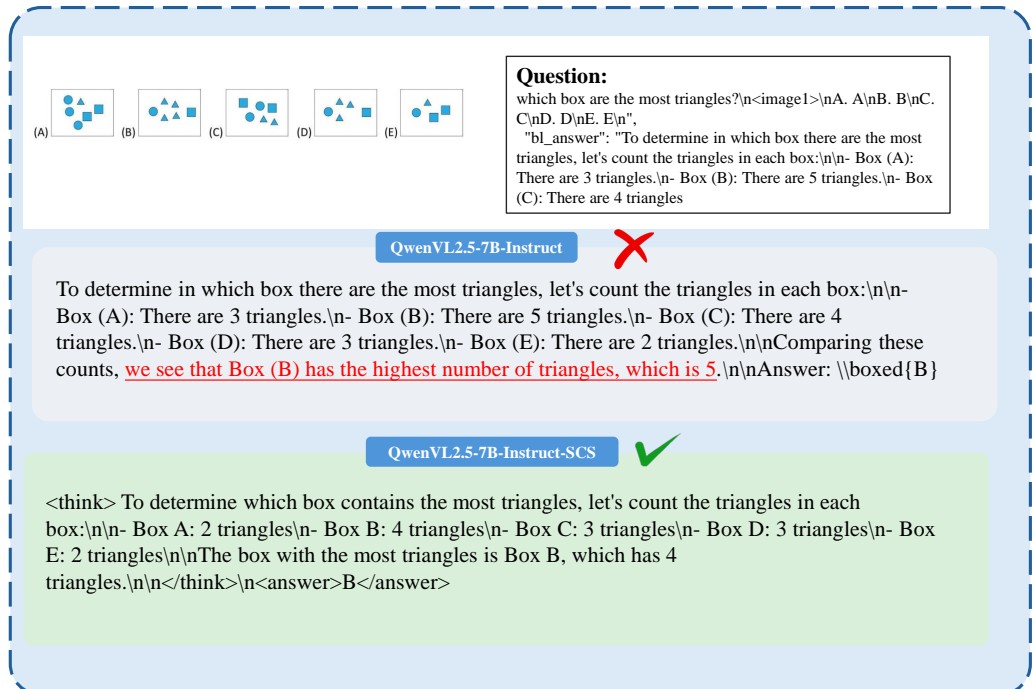

Figure 19: Comparison of model outputs before and after RL training through SCS.

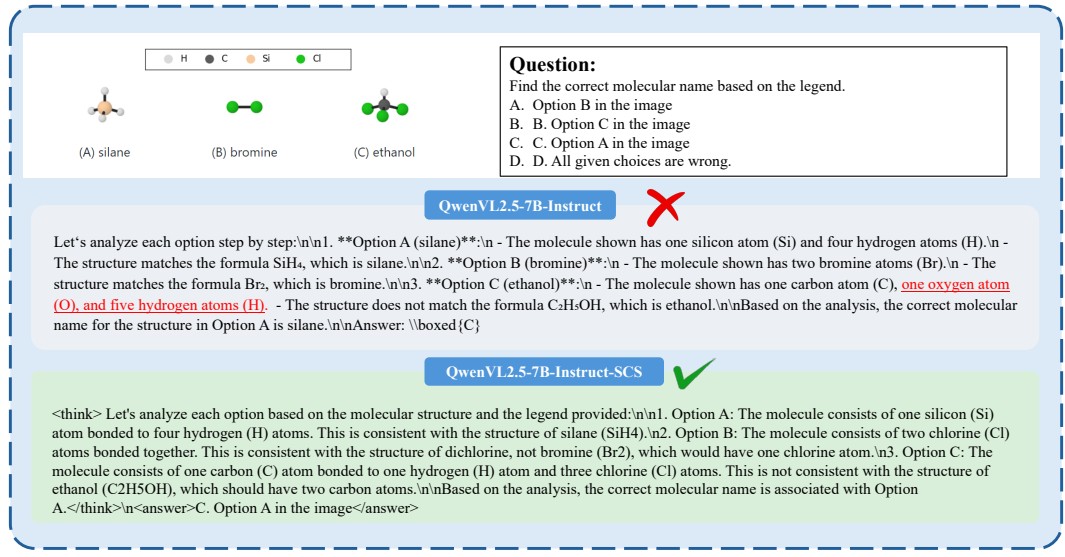

Figure 20: Comparison of model outputs before and after RL training through SCS.

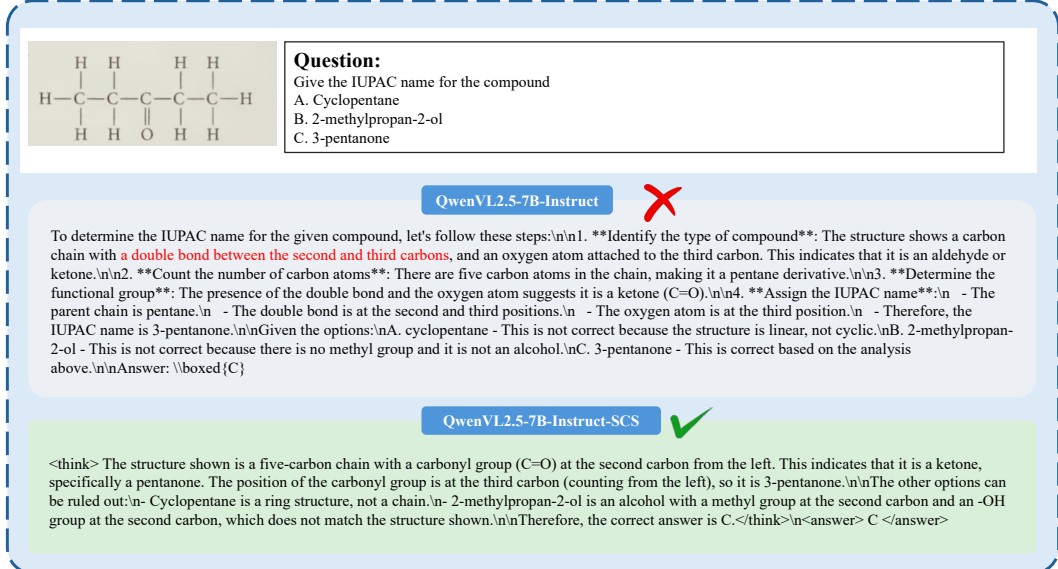

**Question:**
Give the IUPAC name for the compound
A. Cyclopentane
B. 2-methylpropan-2-ol
C. 3-pentanone

**QwenVL2.5-7B-Instruct** ✗

To determine the IUPAC name for the given compound, let's follow these steps:\n\n1. **Identify the type of compound**: The structure shows a carbon chain with a double bond between the second and third carbons, and an oxygen atom attached to the third carbon. This indicates that it is an aldehyde or ketone.\n\n2. **Count the number of carbon atoms**: There are five carbon atoms in the chain, making it a pentane derivative.\n\n3. **Determine the functional group**: The presence of the double bond and the oxygen atom suggests it is a ketone (C=O).\n\n4. **Assign the IUPAC name**:\n - The parent chain is pentane.\n - The double bond is at the second and third positions.\n - The oxygen atom is at the third position.\n - Therefore, the IUPAC name is 3-pentanone.\n\nGiven the options:\nA. cyclopentane - This is not correct because the structure is linear, not cyclic.\nB. 2-methylpropan-2-ol - This is not correct because there is no methyl group and it is not an alcohol.\nC. 3-pentanone - This is correct based on the analysis above.\n\nAnswer: \\boxed{C}

**QwenVL2.5-7B-Instruct-SCS** ✓

<think> The structure shown is a five-carbon chain with a carbonyl group (C=O) at the second carbon from the left. This indicates that it is a ketone, specifically a pentanone. The position of the carbonyl group is at the third carbon (counting from the left), so it is 3-pentanone.\n\nThe other options can be ruled out:\n- Cyclopentane is a ring structure, not a chain.\n- 2-methylpropan-2-ol is an alcohol with a methyl group at the second carbon and an -OH group at the second carbon, which does not match the structure shown.\n\nTherefore, the correct answer is C.</think>\n<answer> C </answer>

Figure 21: Comparison of model outputs before and after RL training through SCS.

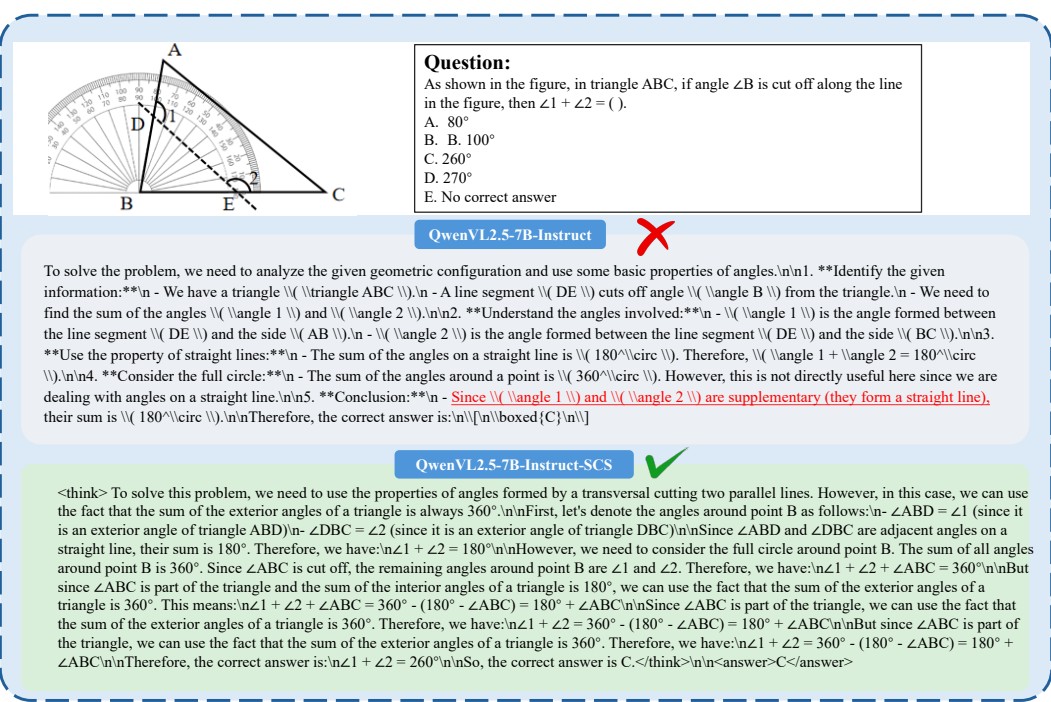

**Question:**
As shown in the figure, in triangle ABC, if angle ∠B is cut off along the line in the figure, then ∠1 + ∠2 = ( ).
A. 80°
B. B. 100°
C. 260°
D. 270°
E. No correct answer

**QwenVL2.5-7B-Instruct** ✗

To solve the problem, we need to analyze the given geometric configuration and use some basic properties of angles.\n\n1. **Identify the given information:**\n - We have a triangle \\( \\triangle ABC \\).\n - A line segment \\( DE \\) cuts off angle \\( \\angle B \\) from the triangle.\n - We need to find the sum of the angles \\( \\angle 1 \\) and \\( \\angle 2 \\).\n\n2. **Understand the angles involved:**\n - \\( \\angle 1 \\) is the angle formed between the line segment \\( DE \\) and the side \\( AB \\).\n - \\( \\angle 2 \\) is the angle formed between the line segment \\( DE \\) and the side \\( BC \\).\n\n3. **Use the property of straight lines:**\n - The sum of the angles on a straight line is \\( 180^\\circ \\). Therefore, \\( \\angle 1 + \\angle 2 = 180^\\circ \\).\n\n4. **Consider the full circle:**\n - The sum of the angles around a point is \\( 360^\\circ \\). However, this is not directly useful here since we are dealing with angles on a straight line.\n\n5. **Conclusion:**\n - Since \\( \\angle 1 \\) and \\( \\angle 2 \\) are supplementary (they form a straight line), their sum is \\( 180^\\circ \\).\n\nTherefore, the correct answer is:\n\\[\n\\boxed{C}\n\\]

**QwenVL2.5-7B-Instruct-SCS** ✓

<think> To solve this problem, we need to use the properties of angles formed by a transversal cutting two parallel lines. However, in this case, we can use the fact that the sum of the exterior angles of a triangle is always 360°.\n\nFirst, let's denote the angles around point B as follows:\n- ∠ABD = ∠1 (since it is an exterior angle of triangle ABD)\n- ∠DBC = ∠2 (since it is an exterior angle of triangle DBC)\n\nSince ∠ABD and ∠DBC are adjacent angles on a straight line, their sum is 180°. Therefore, we have:\n∠1 + ∠2 = 180°\n\nHowever, we need to consider the full circle around point B. The sum of all angles around point B is 360°. Since ∠ABC is cut off, the remaining angles around point B are ∠1 and ∠2. Therefore, we have:\n∠1 + ∠2 + ∠ABC = 360°\n\nBut since ∠ABC is part of the triangle and the sum of the interior angles of a triangle is 180°, we can use the fact that the sum of the exterior angles of a triangle is 360°. This means:\n∠1 + ∠2 + ∠ABC = 360° - (180° - ∠ABC) = 180° + ∠ABC\n\nSince ∠ABC is part of the triangle, we can use the fact that the sum of the exterior angles of a triangle is 360°. Therefore, we have:\n∠1 + ∠2 = 360° - (180° - ∠ABC) = 180° + ∠ABC\n\nBut since ∠ABC is part of the triangle, we can use the fact that the sum of the exterior angles of a triangle is 360°. Therefore, we have:\n∠1 + ∠2 = 360° - (180° - ∠ABC) = 180° + ∠ABC\n\nTherefore, the correct answer is:\n∠1 + ∠2 = 260°\n\nSo, the correct answer is C.</think>\n\n<answer>C</answer>

Figure 22: Comparison of model outputs before and after RL training through SCS.

Table 11: Hyperparameter ablation study for GRPO with SCS.

| r ↓ / m → | 4 | 6 | 8 |
|---|---|---|---|
| **0.2** | 63.6 | 64.1 | 64.4 |
| **0.4** | – | 64.5 | 64.2 |
| **0.8** | 63.2 | – | – |

Table 12: Hyperparameter ablation study for REINFORCE++ with SCS.

| r ↓ / m → | 4 | 6 | 8 |
|---|---|---|---|
| **0.2** | 61.9 | 62.1 | 62.3 |
| **0.4** | 62.7 | – | – |
| **0.8** | 61.1 | – | 62.9 |

Table 13: Hyperparameter ablation study for REINFORCE++–baseline with SCS.

| r ↓ / m → | 4 | 6 | 8 |
|---|---|---|---|
| **0.2** | 63.0 | 63.1 | 62.7 |
| **0.4** | – | – | – |
| **0.8** | 63.0 | – | – |

## D.4 Additional Ablations

We conduct some hyperparameter analysis across all RL methods. ( GRPO, REINFORCE ++ and REINFORCE ++-baseline). The experiment results are shown in Tables 11, 13 and 12:

From the tables, we obtained trends that closely mirror those reported for RLOO:

**GRPO + SCS.** Fixing the ratio (r=0.2), the performance grows as the increase of the the number of resampled trajectories, and finally reaches the peak at 64.4 (r=0.2,m=8).

**REINFORCE++ + SCS.** When fixing the ratio, the same unimodal pattern as GRPO appears. When fixing number of resampled trajectories, the performance metric rises at first and then declines as the ratio increases($61.9 \rightarrow 62.7 \rightarrow 61.1$ for r=0.2,0.4,0.8).

**REINFORCE++–baseline+ SCS.** Accuracy peaks at 63.1 for (r=0.2,m=6). When fixing r=0.2, performance first rises as the number of resampled trajectories grows (63.0 to 63.1 when truncation ratio from 0.2 to 0.4). then slips when the number of resampled trajectories larger (r=0.8,63.2). Nearly all three algorithms every (r,m) configuration still outperforms its vanilla counterpart by 0.6–2.2 pp, indicating that the consistency reward delivers a uniform benefit on different algorithms and hyperparameter settings.

## D.5 Measurement of additional compute costs.

The extra overhead introduced by SCS method lies almost entirely in the sampling phase Leveraging modern high-performance inference engines such as vLLM, these process are batched and run in parallel, so the wall-clock impact grows sub-linearly with N and remains well-controlled. Under identical hyperparameters on 8 * A100 GPU (N=4, truncation ratio=0.8), we observed:

Thus, with advanced inference backends, the time cost of SCS is both predictable and acceptable given the performance gains it enables.

## E Other

All benchmark datasets used for evaluation are properly cited within the manuscript. For all evaluated models, we strictly comply with their respective licenses: open-source models are employed in accordance with their designated usage terms. The training pipeline is implemented based on the

Table 14: Comparison of training time costs after applying SCS.

| Configuration | Training Time | Time Change | Scores | Improvement |
|---|---|---|---|---|
| Baseline | 12.5 h | — | 57.8 | — |
| Baseline + SCS | 17.2 h | $\approx$ +38% | 65.5 | +7.7 |

open-source framework OpenRLHF [*], while the evaluation is conducted using established open-source libraries, including Transformers [†].

**Limitations.** As for limitations, our SCS has not been extensively applied to LLMs and more MLLMs to verify its generality.

**Broader Impact.** We hope this work provides valuable insights into reasoning and supports the continued advancement of MLLMs. Currently, we do not have any ethical or societal risks associated with this research.

---

[*]https://github.com/OpenRLHF/OpenRLHF
[†]https://github.com/huggingface/transformers

