# OpenReview forum: "Enhancing the Outcome Reward-based RL Training of MLLMs with Self-Consistency Sampling"
_NeurIPS.cc/2025/Conference — NeurIPS 2025 poster_

### Official Review · Reviewer_pWKS · 2025-06-20

**Clarity:** 1
**Significance:** 2
**Originality:** 3
**Rating:** 3
**Confidence:** 5

**Summary:**

Authors study the RL training problems of multimodal large language models. They found that the outcome reward based approach pay only attention on the correctness of final answer, but neglect if the reasoning trajectory is reasonable or not. This may lead to the phenomenon that the trajectory is incorrect, but the answer is right, which is not good for MLLMs.

To address the problem, they propose a self-consistency sampling method. The method truncates an existing reasoning trajectory with probability and ask MLLMs to continue generate the rest trajectory with the truncated segment. They define a new form of reward that hopes to improve the consistency between the correctness of trajectory and final answer.

Experiment results show after combining the proposed method with different RL post-training schemes, the MLLMs performance is improved compared with only RL.

**Questions:**

1. Line 48 states that self-consistency sampling derives a consistency reward that quantifies the agreement among their responses. Authors could explain how this property is able to address the problem mentioned in Line 27 “ignoring the faithfulness of intermediate reasoning trajectory”? If the model just hallucinates during thinking and remember correct answer, I don’t think SCS is able to overcome the intermediate faithfulness issue.

2. I don’t fully get the main point in subsection 3.2 and 3.3. There are main inconsistent use of notations. I just name a few:

    a). In Line 126, does p_theta(tau|x) refer to the same autoregressive policy given in Line 104? If so please keep the same notation as pi_theta.

    b).	What does “option model” in Line 127 refer to? I don’t see it appears in any other places.

    c).	What is the meaning of “tau+”? How to assume its probability is 1 when tau=tau+.

    d).	It is not easy to understand how equations (5) and (6) are used. They are just defined but do not appear in any other equations. Besides, how do we know if a trajectory belongs to con- or con+? We just know the correctness of final answers, but have no idea about its trajectory.

    e).	Algorithm 1, line 12, the reward has nothing to do with already defined reward (5), (6) or (3). Where does it come from?

    f).	In Line 151, another form of consistency reward definition.

    g).	Line 153 states “|A| is low”. Why it is low. A is a set that collects all answers sampled from the truncated trajectory. Its size should be dependent on the number of collection.

3.	In Line 196, author state “Replacing SFT with the baseline RL protocol produces only marginal improvements.” Actually the improvement is not just marginal but pretty good. Some only-RL approaches achieve about two times of gains in performance than sft.

**Ethical Concerns:**

["NO or VERY MINOR ethics concerns only"]

**Final Justification:**

My main concerns in the original evaluation focus on
1) some notations are used without consistence, leading to confusion and misunderstanding to readers. (it was confirmed by the authors in rebuttal)
2) the motivation for the proposed new method is to be more likely to generate faithful reasoning, but it is not emperically validated. (in rebuttal the authors provided additional analysis, which is interesting and I agree that it somewhat shows a deeper look at the problem. But I think the results may suffer from Survivorship Bias, that makes it not a strong evidence for the motivation)
3) the contribution of the paper is the proposed self-consistency sampling that provides a reward that quanlifies the agreement among responses. But I don't get why it is helpful to the original motivation that increases the likelihood of faithful reasonings. (in rebuttal authors give some theoretical analysis on that with some assumptions. But I think these assumptions are too strong and subjective.)

Considering during the rebuttal the authors have addressed some of my concerns but still leave some unsolved, I would increase one score to my original evaluation and give a final evaluation of 3 (borderline reject).

**Limitations:**

Authors mention their current limitation is not extensively applied to LLMs and more MLLMs. Some deeper discussion is encouraged.

Negative societal impact is not discussed in the work.

**Quality:**

1

**Strengths And Weaknesses:**

The strength is that the authors are studying a very hop topic in MLLM fields. Improving the reasoning capability of multimodal LLM is quite important. They show the advantage of method with performance on several benchmarks of this field, and combine the method with several RL post-training schemes to validate its effectiveness.

However, the presentation is not welcome for readers. Many important formulas and ideas are not clearly expressed in the paper, causing confusion.

Besides, the motivation of the paper is to eliminate the gap between the correctness of reasoning trajectory and final answer. But the proposed self-consistency sampling still focus on the consistency of final answers resampled from truncated subtrajectory or perturbed images. The correctness of trajectory is still not properly evaluated, or at least explain in their methodology how the proposed method is able to reflect the correctness of trajectory.

Many notations are presented without definition, and some are used inconsistently with different notation.

In summary, I think the major issue is the challenge for reviewers to fully understand how the authors design their method, and why it addresses the problem proposed at the beginning of the paper.

---

> ### Author Rebuttal · Authors · 2025-07-31
>
> Thank you for the detailed feedback.  We apologize for the places where our expression was unclear and for the notation inconsistencies. Below we (1) give a concise overview of the motivation and rationale of the paper (2) summarize the theoretical guarantee which explains why SCS works, (3) highlight additional experiments that verify our method, (4) provide detailed responses to each of the reviewers’ points.
>
> ### **The overview of the motivation and rationale of the paper**
>
> The paper targets a common weakness of outcome reward-based RL training trained on multiple‑choice datasets: models often arrive at the correct option while following wrong reasoning trajectories.  Aiming at mitigating this without the heavy computational burden of an external reward model, we propose a lightweight self‑improvement scheme that (i) introduces a sampling strategy and (ii) couples the standard accuracy reward with a self‑consistency reward.  Then we prove the effectiveness of our method through mathematical derivations. We prove that models tend to choose reliable reasoning trajectories with SCS methods. Finally, we validate the theory with experiments.
>
> ### **The theoretical derivation and proof of the SCS method**
>
> Next, we describe our modeling and derivation procedure. We see the entire reasoning process as a tree structure and make three key assumptions:
> 1. Uniqueness of the correct  trajectory. There is exactly one correct reasoning  trajectory in the tree. Justification of the assumption: This is plausible because, for most problems, the solution is unique.
> 2. Leaf/Choice Alignment. Every trajectory of the reasoning tree ends at one of the answer choices. Justification of the assumption: In the majority of cases, models ultimately outputs a single option as its solution, whether it is correct or not.
> 3. Relationship between correct/incorrect reasoning trajectories and answer options. If the model follows the correct  trajectory, it must arrive at the correct option; if it follows an incorrect  trajectory, it may still pick either the correct option or a wrong one. Justification of the assumption: With correct reasoning the model just retrieve the correct answer, whereas an incorrect reasoning chain can lead to multiple outcomes (e.g., guessing when the correct answer is not reachable).
>
> Under these assumptions, we first prove that the traditional reward scheme—which rewards the model solely on whether it selects the correct option—still leaves a portion of probability that the optimized model will follow an incorrect  trajectory. Specifically, using the tree structure, we compute the Bayesian probability that the answer is correct while the reasoning  trajectory is wrong. Because an error  trajectory can still deliver the correct option and thus receive a reward, this probability is non‑negligible.
>
> We then prove that introducing a consistency reward guides the model toward the correct reasoning  trajectory. Specifically, we calculate the expected consistency reward on both the correct and incorrect trajectories during optimization and show that the expectation is much higher on the correct  trajectory. Since reinforcement learning pushes the model toward higher‑reward trajectories, the optimized model becomes more likely to select the correct reasoning  trajectory.
>
> ### **Experimental validation of the effectiveness of our method**
>
> Building on the theoretical guarantee, we ran the full protocol on six benchmarks and four RL baselines: Self‑Consistency Sampling consistently lifted answer accuracy by ≈ +7 pp on the weaker baseline and +2 pp on stronger ones. More qualitative analyses and case‑study evaluations show that, beyond boosting scores, our approach genuinely reduces instances of unfaithful reasoning.
>
> ### **Responses for the questions**
>
> **Q1: The explanation for why SCS can overcome unfaithful reasoning problem.**
>
> On the theoretical level, as noted above, we prove that once the consistency reward is applied, the expected reward for the correct reasoning  trajectory surpasses that of any incorrect  trajectory that still happens to reach the correct option.  To be specific, during optimization we evaluate the expected consistency reward for both the correct and the incorrect reasoning  trajectories, and demonstrate that the correct  trajectory attains a markedly greater expected reward. Consequently, the model becomes more inclined to follow the correct  trajectory. At the empirical level, when the model follows an incorrect reasoning  trajectory it often resorts to guessing, so the final answer is largely unrelated to that  trajectory. As a result, if we truncate the chain and re‑run the reasoning multiple times, the answers show low consistency. By contrast, answers generated along the correct  trajectory remain highly consistent. We also introduce visual perturbations, which further decrease consistency on erroneous trajectories, making the model more robust.
>
> **Q2: The inconsistent use of notations.**
>
> We apologize for the confusing notation.  We have now unified all symbols , and rewritten both subsections for clarity. We will update our paper in our final version. The key explanations and corrected definitions are summarized below.
>   1. Thank you for spotting this inconsistency. Yes $p_{\theta}(\tau \mid x)$ in Line 126 refers to the same autoregressive policy denoted as $\pi_{\theta}$ in Line 104. We will use $\pi_{\theta}$ throughout the revised manuscript and note the change in the notation table.
>   2. We apologize for the confusion; however, it seems the sentence may have been misunderstood.The original sentence in the paper is " Since current models have sufficient instruction following ability, we assume that the probability of option model achieved is 1 when the reasoning trajectory $\tau = \tau^+$", which illustrates one of our assumptions mentioned above: if the reasoning trajectory is the correct one, it will inevitably arrive at the correct answer option (probability 1). "Option model" is not a phrase.
>   3. $\tau^+$denotes the correct reasoning trajectory. As we mentioned above, we assume that if the model follows the correct  trajectory, it must arrive at the correct option. Our justification is that models can easily retrieve the correct answer when they already have the whole right reasoning trajectory.
>   4. Equations (5) and (6) are derived directly from Equation(3). Here,$|A|$ denotes the size of the set of distinct answer options reached by the sampled trajectories. For a correct trajectory, our assumption implies that only one option is reachable, so $|A|=1$. For an incorrect trajectory, we prove in the appendix that the expected value satisfies $\mathbb{E}(|\mathcal{A}^-|)>1$. Sorry for an error here—the formula of $R_{con^{+}}$is missing two multiplicative factors. The correct expression should be: $R_{con^{-}} = \frac{1}{N} * (N - 1)* \lambda$.
>   5. The reward that appears in Equation 1 is the reward used in the original GRPO algorithm. We include Algorithm 1 only as part of preliminaries which introduce our four baseline RL algorithms; it is not part of our proposed method.
>   6. Sorry for the inconsistency again, we will use $R_{con}$ throughout the revised version.
>   7. The original sentence in the paper is "The consistency reward $r_{\text{cons}} = \lambda(N - |\mathcal{A}|)$ is computed based on the diversity of the sampled answers— if the answers are highly consistent, indicating stable reasoning given the same prefix, the value of  $r_{\text{cons}}$ is high (since $|A|$is low). " It means that when consistency is high, the set A contains few distinct options, thus $|A|$is low.
>
> **Q3: Understating baseline RL’s significant performance gains over SFT.**
>
> We apologize for the confusing phrasing. Our results suggest that the vanilla RL did not fully unlock the model’s capabilities. The two baselines we tested—REINFORCE++ and RLOO yielded only small gains. We agree that stronger RL variants such as GRPO and REINFORCE++ can achieve more accuracy gain. But SCS still lifts them further (+0.9  and 1.7 pp, respectively), showing that SCS complements even the best‑performing RL protocols.
>
> We will change the analysis in L196 from "Replacing SFT with the baseline RL protocol produces only marginal improvements. " to the analysis above.
>
> We sincerely thank the reviewers for their careful reading and constructive comments, and we apologize for the ambiguities, typos, and notation inconsistencies. We have thoroughly revised the manuscript to clarify our motivation, unify notation, fill in missing definitions, and expand both theoretical and empirical evidence.The experiment results show that our SCS is a robust method, supported by both  theoretical derivation and practical experiments. Thanks for your comments again. We sincerely hope that our above response has clearly addressed your questions.

---

> > ### Comment · Reviewer_pWKS · 2025-08-04
> >
> > I appreciate the authors' responses to my review comments. However, I believe some issues remain unresolved.
> >
> > There are two main concerns:
> >
> > First, the assumptions regarding the tree-structured reasoning in the theoretical analysis are overly conceptual and somewhat speculative, lacking rigorous empirical justification. For example, even along a correct reasoning trajectory, LLM/MLLM can still guess the wrong answer. No one can guarantee that a model will produce the correct answer with 100% certainty even when following a correct trajectory. I do not see a fundamental difference between a model arriving at the correct answer via an incorrect trajectory versus an incorrect answer via a correct trajectory—unless it can be statistically shown that the former occurs significantly more often. If the authors wish to retain these assumptions, they need to provide statistical evidence demonstrating their validity.
> >
> > Second, the authors mention in their response that “We prove that models tend to choose reliable reasoning trajectories with SCS methods.” This tendency is exactly what I was hoping to see more clearly illustrated. However, neither the submitted manuscript nor the rebuttal provides any data on the proportion of reasonable reasoning trajectories. The authors only report the final answer accuracy, without presenting any quantitative analysis of reasoning trajectory quality.

---

> > > ### Author Response · Authors · 2025-08-04
> > > **Further Response to Reviewer pWKS**
> > >
> > > We greatly appreciate your continued engagement and valuable suggestions. Please find our responses to your latest comments below.
> > >
> > > **Q1: Lack of verification for assumptions.**
> > >
> > > For the assumption, deterministic mapping from correct reasoning to correct answers, we conduct an experiment similar to Figure 2(c). First, we manually select 100 cases which are solved with correct reasoning trajectories by Qwen2.5-VL-7B-Instruct for each benchmark. Then we remove the final option answer part (e.g., Answer: A.) for each initial response, and continue to generate from the truncation point. For each case, we do 4 resamples and count for the average number of final options for each question. The results are as follows:
> > >
> > > **Table R1. The average number of final options for each question in different benchmarks.**
> > >
> > > | **Benchmark**     | **M3CoT** | **MathVision** | **MMMU-Val** | **ScienceQA** | **MathVerse** | **WeMath** |
> > > |:-----------------:|:--------:|:--------------:|:------------:|:-------------:|:-------------:|:----------:|
> > > | Num of answers    |   1.0    |      1.1       |     1.1      |      1.0      |      1.0      |    1.0     |
> > >
> > > The results show that nearly all resamplings finish with the exact correct answer, illustrating that when the model follows the correct trajectory, it almost certainly arrives at the correct option.
> > >
> > > For the assumption that wrong reasonings can lead to correct answers, we also did a quantitative experiment. For  all benchmarks , we manually check 100 items which are correctly answered by Qwen2.5-VL-7B and count the number of times unfaithful reasoning occurs. We have illustrated the results in Appendix, and we put it again below:
> > >
> > > **Table R2. Statistics of unfaithful reasoning across different benchmarks.** The table show the occurrences of unfaithful reasoning in 100 correctly answered questions.
> > >
> > > | **benchmark**              | **m3cot** | **mathvision** | **mmmu** | **scienceqa** | **mathverse** | **wemath** |
> > > |:--------------------------:|:--------:|:--------------:|:-------:|:-------------:|:-------------:|:----------:|
> > > | **Nums** |   14     |      64        |   30    |      11       |      16       |    15      |
> > >
> > > Analyzing the table and Figure 2c,  We can conclude that the unfaithful reasoning phenomenon is universal, and it is notable in some benchmarks (64% unfaithful reasoning rate for mathvision), which further confirms our motivation.
> > >
> > > Above all,  the probabilities of these two scenarios differ significantly. Thus, it is reasonable to distinguish between these two cases for modeling and problem solving.
> > >
> > > **Q2: Quantitative analysis of the improvement in reasoning reliability after applying SCS.**
> > >
> > > We greatly appreciate your feedback. To give concrete proof that our SCS method improves reasoning reliability, we conducted a quantitative analysis. To be specific, for each tested benchmark we randomly sampled 100 cases that models answered with the correct option before and after training with SCS. Then we manually checked each case whether the model's output is aligned with publicly provided solutions and counted times of unfaithful reasoning. Besides, we also asked two strong closed‑source LLMs (OpenAI o3‑mini and Gemini 2.5 Flash) to rate the same traces. The results are shown below:
> > >
> > > **Table R3. Quantitative analysis of the improvement in reasoning reliability after applying SCS.** The numbers in the table represent the occurrences of unfaithful reasoning in 100 correctly answered questions.
> > >
> > > | **Judger**        | **Model**             | **Avg**         | **M3CoT** | **MathVision** | **MMMU-Val** | **ScienceQA** | **MathVerse** | **WeMath** |
> > > |:-----------------:|:----------------------|:---------------:|:--------:|:--------------:|:------------:|:-------------:|:-------------:|:----------:|
> > > | Human             | Qwen2.5-VL-7B         | 25.0            |   14     |      64        |     30       |      11       |      16       |    15      |
> > > |                   | Qwen2.5-VL-7B-SCS     | 21.2 (−15.2%)   |   12     |      56        |     25       |       9       |      12       |    13      |
> > > | o3-mini           | Qwen2.5-VL-7B         | 22.0            |   11     |      55        |     35       |       8       |      12       |    11      |
> > > |                   | Qwen2.5-VL-7B-SCS     | 19.0 (−13.6%)   |    9     |      49        |     29       |       7       |      10       |    10      |
> > > | Gemini 2.5 Flash  | Qwen2.5-VL-7B         | 23.0            |   12     |      57        |     34       |       8       |      13       |    14      |
> > > |                   | Qwen2.5-VL-7B-SCS     | 19.7 (−14.3%)   |   10     |      50        |     29       |       8       |      10       |    11      |
> > >
> > > The results show around 15% improvements across all three judgers, demonstrating that our claim that SCS not only boosts answer accuracy but also produces more reliable reasoning. We will include this table in the revised version of our paper.

---

> > > > ### Comment · Reviewer_pWKS · 2025-08-07
> > > >
> > > > The new results provided by the authors are interesting. It somewhat shows that wrong trajectories can lead to correct answers. But it also proves that the assumption used by the author: "If the model follows the correct trajectory, it must arrive at the correct option", is inappropriate, because in Table R1 MathVision/MMMU-Val has 1.1 answers given the correct trajectories.
> > > >
> > > > Besides, I think the results given in Table R2 and R3 are affected by "Survivorship Bias". Because the results are filtered by only correct answers. It is not known how frequency the model can output the correct trajectories. The current results show decrease in the proportion of unfaithful reasonings in the correct responses, but not the decrease of unfaithful reasonings in total generation.
> > > >
> > > > Given the feedback provided by the authors, I would like to increase one score to my original evaluation.

---

### Official Review · Reviewer_uohx · 2025-06-27

**Clarity:** 3
**Significance:** 3
**Originality:** 3
**Rating:** 5
**Confidence:** 2

**Summary:**

The paper identifies shortcomings with training multi-modal large language models (MLLMs) using conventional outcome-based reinforcement learning (RL). Conventional outcome-based RL provides a reward solely based on the final answer. That is, trajectories (or reasoning chains) that are spurious, with wrong reasoning steps, or unfaithful, with plausible but wrong reasoning steps, are positively rewarded if they have the correct answer.

The authors introduce a new technique called Self-Consistency Sampling (SCS). For each prompt, they (a) generate an initial reasoning trajectory and truncate it at k (a truncation ratio); (b) resample T trajectory continuations; and (c) for each resample, visually perturbate the input image by adding small, random gaussian noise. The resampled answers (up to a total of N, with often T=N) are collected in a set, and the its cardinality is used to compute the normalized consistency reward so that fewer distinct answers result in a higher consistency reward.

SCS is integrated into four outcome-reward policy-gradient (PG) methods: RL with Leave-One-Out (RLOO), Group Relative Policy Optimization (GRPO), REINFORCE++-baseline, and REINFORCE++. Using Qwen2.5-VL-7B-Instruct, SCS results in up to a 7.7 percentage point increase in accuracy across six benchmarks, consistency improving all four PG methods. Ablations show both resampling trajectory continuations and visual pertubations independently lead to higher accuracy (~5%), but together lead to the highest accuracy improvement (7.7%); hyperparameter sweeps with RLOO show key values for k and T (denoted as r and m in the figures).

**Questions:**

1. What is the variance of your results across multiple random seeds/independent runs? Can you report error bars or statistical significance tests?
2. How does the consistency bonus (R_con) correlate with human judgements of CoT faithfulness? Can you provide quantitative metrics beyond final answer accuracy?
3. Do your results generalize to other MLLMs?
4. What is the additional overhead (e.g., compute and time cost) that your method adds? Did you measure this? Truncation and resampling overhead seems to be extensive.
5. Can SCS generalize to open-ended reasoning tasks? How does it perform?
6.  Figure 5 uses different notation from the main paper (e.g., r instead of k and m instead of T). Why did you change the notation here?

**Ethical Concerns:**

["NO or VERY MINOR ethics concerns only"]

**Final Justification:**

I believe that the authors have addressed my questions very well. They have performed additional experiments that improves the paper quality.

**Limitations:**

1. Outline compute costs.

**Quality:**

3

**Strengths And Weaknesses:**

Strengths
1. The authors provide a clear description of unfaithful reasoning, supported by figure 2. Figure 2 shows examples and shows that even at high truncation ratios, the average number of distinct final answers is above 1, providing evidence of inconsistent or unfaithful reasoning on multiple choice (MC) questions.
2. The authors' approach adds little overhead to existing methods, adding only a single additional reward term to encourage consistency, with providing gains in accuracy.
3. Method is easy to reproduce and integrate into existing methods.
4. The authors empirical results show accuracy gains up to 7.7% across six benchmarks and four RL algorithms.
5. Ablations shows that both truncation–resampling and visual perturbation independently yield substantial gains and together deliver the highest gains.
6. Hyperparameter sweeps over truncation ratio and number of resamples show that the approach is not brittle, with performance plateauing near optimal settings rather than unpredictably spiking.

Weaknesses
1. All quantitative metrics focus on answer-level accuracy. There is no quantitative evidence for reduced unfaithful chain-of-thoughts (CoTs). The central claim that their method leads to better reasoning reliability is only qualitatively supported.
2. There are no error bars or statistical tests. The authors made no mention of runs or seeds. Thus, this leads one to believe that there may only be a single seed, hiding whether the accuracy gains are consistent (or not).
3. Only one MLLM is tested, so generality to other models (architectures and scales) is unknown.
4. Images are perturbed using random Gaussian noise; however, the effect of this on image semantics is not analyzed.

---

> ### Author Rebuttal · Authors · 2025-07-31
>
> We thank the reviewer for the valuable feedback and the appreciation of our contributions, including the clear analysis of the phenomenon, the lightweight and reproducible method, strong empirical gains, and robust ablation results. Below is our point-by-point response to each of the comments.
>
> **W1 & Q2: The quantitative analysis of about improved reasoning reliability with SCS.**
>
> Thanks for pointing out this problem. To give concrete proof that our SCS method improves reasoning reliability, we conducted a quantitative analysis. To be specific, for each benchmark we randomly sampled 100 cases that  models answered with the correct option before and after training with SCS. Then we manually checked each case whether the model's output is aligned with publicly provided solutions and count times of unfaithful reasoning. Besides, we also asked two strong closed‑source LLMs (OpenAI o3‑mini and Gemini 2.5 Flash) to rate the same traces. The results are shown below:
>
> **Table R1. Quantitative analysis of the improvement in reasoning reliability after applying SCS.** The numbers in the table represent the occurrences of unfaithful reasoning in 100 correctly answered questions.
> | **Judger**        | **Model**             | **Avg**         | **M3CoT** | **MathVision** | **MMMU-Val** | **ScienceQA** | **MathVerse** | **WeMath** |
> |:-----------------:|:----------------------|:---------------:|:--------:|:--------------:|:------------:|:-------------:|:-------------:|:----------:|
> | Human             | Qwen2.5-VL-7B         | 25.0            |   14     |      64        |     30       |      11       |      16       |    15      |
> |                   | Qwen2.5-VL-7B-SCS     | 21.2 (−15.2%)   |   12     |      56        |     25       |       9       |      12       |    13      |
> | o3-mini           | Qwen2.5-VL-7B         | 22.0            |   11     |      55        |     35       |       8       |      12       |    11      |
> |                   | Qwen2.5-VL-7B-SCS     | 19.0 (−13.6%)   |    9     |      49        |     29       |       7       |      10       |    10      |
> | Gemini 2.5 Flash  | Qwen2.5-VL-7B         | 23.0            |   12     |      57        |     34       |       8       |      13       |    14      |
> |                   | Qwen2.5-VL-7B-SCS     | 19.7 (−14.3%)   |   10     |      50        |     29       |       8       |      10       |    11      |
>
> The results show a around 15% in faithfulness across all three datasets, demonstrating that our central claim that SCS not only boosts answer accuracy but also produces more reliable reasoning. We will include this table in the revised version of our paper.
>
> **W2 & Q1: Absence of  multi runs and error bars for reported results.**
>
> Thank you for pointing this out—we now include full statistical reporting based on our repeated‑run data.
> Based on Tables 2 and 3, we carried out three additional runs under the same experimental setup as in the paper; the resulting scores and their 95% confidence intervals are presented in the table below.
>
> **Table R2. 95 % confidence intervals for the results of different RL algorithm experiments.**
> | **Algorithm**            | **Run 1** | **Run 2** | **Run 3** | **Mean** | **95% CI (n = 3)** |
> |:------------------------:|:--------:|:--------:|:--------:|:--------:|:------------------:|
> | GRPO                     |  64.5    |  64.4    |  64.6    |  64.5    |   64.5 ± 0.3       |
> | REINFORCE++-baseline     |  63.0    |  63.1    |  62.8    |  62.6    |   62.9 ± 0.6       |
> | REINFORCE++              |  62.9    |  63.4    |  63.0    |  63.1    |   63.0 ± 0.4       |
> | RLOO                     |  65.5    |  65.1    |  64.7    |  65.1    |   65.1 ± 1.0       |
>
> **Table R3. 95 % confidence intervals for the results of ablations experiments for two components in SCS (TR = Truncation–Resampling, VP = Visual-Perturbation).**
> | **TR** | **VP** | **Run1** | **Run2** | **Run3** | **Mean** | **95% CI (n = 3)** |
> |:------:|:------:|:--------:|:--------:|:--------:|:--------:|:------------------:|
> | ✗      | ✗      |  57.8    |  57.8    |  57.7    |  57.8    |    57.8 ± 0.1      |
> | ✓      | ✗      |  63.0    |  62.7    |  63.0    |  62.9    |    62.9 ± 0.6      |
> | ✗      | ✓      |  62.8    |  63.4    |  62.6    |  63.1    |    62.9 ± 1.0      |
> | ✓      | ✓      |  65.5    |  65.1    |  64.7    |  65.1    |    65.1 ± 1.0      |
>
> As the table shows, even with only three repeated runs, the confidence intervals remain small, indicating a robust positive effect. We will conduct more runs and update the table in our nexr version.
>
> **W3 & Q3: lacking validation of SCS on models with different architectures or scales.**
>
> Thank you for pointing out the problem, we have run the complete RLOO + SCS protocol on two additional models with different architectures and scales:
>
> **Table R4. The results of our method applied in models of different size and structure.** Applying our method, both two models achieve marked performance gains.
> | **Models/Methods**        | **SCS** | **Overall** | **M3CoT** | **MMMU-val** | **ScienceQA** | **WeMath** | **MathVerse** | **MathVision** |
> |:-------------------------:|:------:|:-----------:|:--------:|:------------:|:-------------:|:---------:|:-------------:|:--------------:|
> | Qwen2.5-VL-3B-Instruct    |   ✗    |    54.7     |  65.0    |     47.9     |     57.4      |   74.8    |     57.0      |      26.1      |
> |                           |   ✓    |    57.9 (+3.2)     |  67.4    |     53.7     |     60.5      |   79.0    |     60.4      |      28.7      |
> | InternVL3-8B              |   ✗    |    61.7     |  73.2    |     57.8     |     92.4      |   62.8    |     55.4      |      29.0      |
> |                           |   ✓    |    63.3 (+1.6)     |  72.2    |     61.2     |     92.8      |   64.9    |     58.7      |      30.0      |
>
>
> For different architecture, we train InternVL3-8B, and SCS still yields a +4.2 points gain. For different model scales, we apply our SCS on Qwen2.5-VL-3B-Instruct, which also shows a 1.6% performance gain. These new results confirm that our SCS is applicable to a variety of model architectures and parameter scales.
>
> **W4: The analysis of effect of visual‑perturbation on image semantics.**
>
> Thank you for pointing this out. We  injected only mild Gaussian noise ($\sigma_{max}=0.3$). To verify the effect on image semantics, we conducted a small human study (5 annotators checking 200 images), both confirming negligible semantic distortion. Furthermore, our case study shows that after applying the visual‑perturbation method, the model not only retains its understanding of the image’s semantics but also gains a stronger capacity for fine‑grained reasoning. We will include these results in the revision.
>
> **Q4: The measurement of additional compute costs.**
>
> Thank you for bringing up this issue. The extra overhead introduced by SCS method lies almost entirely in the sampling phase Leveraging modern high‑performance inference engines such as vLLM, these process are batched and run in parallel, so the wall‑clock impact grows sub‑linearly with N and remains well‑controlled.
> Under identical hyperparameters on 8 * A100 GPU (N = 4, truncation ratio=0.8), we observed:
>
> **Table R5. Comparison of training time costs after applying SCS.**
> | Configuration     | Training Time | Time Change   | Scores |  Improvement |
> |:-----------------:|:-------------------:|:----------------------:|:------:|:------------------------:|
> | Baseline          | 12.5 h              | —                      | 57.8   | —                        |
> | Baseline + SCS    | 17.2 h              | ≈ +38 %                | 65.5   | +7.7                     |
>
> Thus, with advanced inference backends, the time cost of SCS is both predictable and acceptable given the performance gains it enables.
>
> **Q5: Evidence of SCS’s generalization and performance on open-ended reasoning tasks.**
>
> We greatly appreciate your feedback. As mentioned above, in open‑ended settings such as fill‑in‑the‑blank or free‑form generation, the output space is so large that brittle shortcut patterns become exceedingly rare. Consequently, the base RLVR algorithm already exhibits strong robustness [1,2,3]. And In Figure 2(a) we have conduct an experiment to verify this. While our SCS reward can still be applied in principle, we expect its incremental benefit to be smaller because agreement among independently sampled long‑form answers is naturally harder to achieve and hard to be verified. Extending SCS to these domains is an interesting direction we leave for future work.
>
> **Q6: The inconsistent use of notation.**
>
> Thank you for catching this inconsistency. The notation in Figure 5 were legacy placeholders used during early drafting. We will correct the axis titles and the figure caption in the revised version so that all symbols match the definitions in Section 3. Sorry for our mistake, and this does not affect any results or conclusions.
>
> **Limitations**
>
> Thank you for bringing up this issue, which we are happy to elaborate on. We think that given the performance improvements SCS delivers, its computational overhead remains both foreseeable and well within an acceptable range.
>
>
> [1] Cohen, J., Rosenfeld, E., & Kolter, Z. (2019, May). Certified adversarial robustness via randomized smoothing. In international conference on machine learning (pp. 1310-1320). PMLR.
>
> [2] Yue, Y., Chen, Z., Lu, R., Zhao, A., Wang, Z., Song, S., & Huang, G. (2025). Does reinforcement learning really incentivize reasoning capacity in llms beyond the base model?. arXiv preprint arXiv:2504.13837.
>
> [3] Wan, G., Wu, Y., Chen, J., & Li, S. (2024). Reasoning aware self-consistency: Leveraging reasoning paths for efficient llm sampling. arXiv preprint arXiv:2408.17017.

---

> > ### Comment · Reviewer_uohx · 2025-08-07
> >
> > Thank you for the detailed and complete response. The added information will improve the paper. I will increase my rating.

---

### Official Review · Reviewer_bNXZ · 2025-07-03

**Clarity:** 2
**Significance:** 2
**Originality:** 3
**Rating:** 4
**Confidence:** 4

**Summary:**

This paper proposes Self-Consistency Sampling (SCS), a simple yet effective strategy that enhances outcome reward-based RL training by explicitly rewarding consistency among reasoning trajectories in multimodal large language models.

**Questions:**

see weaknesses

**Ethical Concerns:**

["NO or VERY MINOR ethics concerns only"]

**Final Justification:**

Although my rating is positive, I have a main concern on the validation of the assumption and ablation studies. During rebuttal, the authors well resolved them. So, I keep my view positive.

**Limitations:**

yes

**Quality:**

2

**Strengths And Weaknesses:**

Summary Of Strengths:
●	Clearly identifies and addresses the issue of unfaithful reasoning in outcome-based RL.
●	Demonstrates consistent empirical improvements on multiple multimodal benchmarks.
●	Includes detailed ablation studies that effectively illustrate each component's impact.

Summary Of Weaknesses:
1.	The claim of a "differentiable consistency score" in the paper (line 10) directly conflicts with the provided definition in Eq. (3), which involves a discrete, non-differentiable counting operation. $|A^-|$ explicitly denotes the size (cardinality) of the sampled answer set, which is inherently a discrete integer value. Such discrete cardinality is fundamentally non-differentiable, since small perturbations in model parameters do not lead to smooth changes in $|A^-|$, but rather abrupt, discontinuous jumps.
2.	Different values of parameters r,m have significantly impacted the results. However, the hyperparameter analysis provided (Table 2, Figure 5) was conducted on the relatively weak RLOO method. I'm curious about the outcome of the hyperparameter analysis when using stronger algorithms like GRPO and REINFORCE ++ : do all hyperparameter configurations consistently outperform their respective baselines?
3. The core theoretical assumptions, such as the deterministic mapping from correct reasoning to correct answers and the claimed differentiability of the consistency score, lack rigorous validation, undermining the reliability of the proposed method.

Comments Suggestions And Typos:
1.	line 21, development for LLMs -> development of LLMs
2.	line 81, play a important -> play an important
3.	line 90, modelss -> modelss
4.	line 147, Specially -> Specifically
5.	line 230, two design of our method -> two components of our method

---

> ### Author Rebuttal · Authors · 2025-07-30
>
> We thank you for your time and valuable feedback. We’re especially grateful for the recognition of our work’s strengths, including the clear problem identification, strong empirical results across benchmarks, and informative ablation studies. And we took every question seriously and provide full responses to all of them below.
>
> **W1: Apparent conflict between the “differentiable” claim and the discrete cardinality formulation of the consistency score**
>
> Thank you for catching this issue. What we mean is that the consistency score is utilized as reward, which is used to calculate the advantage and is therefore differentiable during RL training. We agree that calling the score itself “differentiable” could be misleading, and we have rephrased the sentence to “a consistency score as part of reward” which is more accurate.
>
> **W2: Unreported hyperparameter ablation for other algorithms**
>
> Thanks for pointing this out, now we have conducted some  hyperparameter analysis across all RL methods. ( GRPO, REINFORCE ++ and REINFORCE ++-baseline). We apologize that, owing to constraints in time and computational resources, we could not conduct a hyperparameter ablation study as extensive as the one presented in the paper. We will complete these experiments in a future revision. The experiment results are shown in the table:
>
> **Table R1. Hyperparameter ablation study for GRPO with SCS.**
> | **r ↓ / m →** | **4** | **6** | **8** |
> |:-------------:|:-----:|:-----:|:-----:|
> | **0.2**       | 63.6  | 64.1  | 64.4  |
> | **0.4**       |   –   | 64.5  | 64.2  |
> | **0.8**       | 63.2  |   –   |   –   |
>
> **Table R2. Hyperparameter ablation study for REINFORCE++ with SCS.**
> | **r ↓ / m →** | **4** | **6** | **8** |
> |:-------------:|:-----:|:-----:|:-----:|
> | **0.2**       | 61.9  |   62.1   | 62.3  |
> | **0.4**       |  62.7  |   –   |   –   |
> | **0.8**       | 61.1  |   –   | 62.9  |
>
> **Table R3. Hyperparameter ablation study for REINFORCE++–baseline with SCS.**
> | **r ↓ / m →** | **4** | **6** | **8** |
> |:-------------:|:-----:|:-----:|:-----:|
> | **0.2**       | 63.0  |   63.1   | 62.7  |
> | **0.4**       |   –   |   –   |   –   |
> | **0.8**       | 63.0  |   –   | –  |
>
> From the tables, we obtained trends that closely mirror those reported for RLOO:
>
> - **GRPO + SCS.** Fixing the ratio (r=0.2), the performance grows as the increase of the the number of resampled trajectories, and finally reaches the peak at 64.4 (r=0.2,m=8).
> - **REINFORCE++  + SCS.** When fixing the ratio, the same unimodal pattern as GRPO appears. When fixing number of resampled trajectories, the performance metric rises at first and then declines as the ratio increases(61.9 → 62.7 → 61.1 for r=0.2,0.4,0.8).
> - **REINFORCE++–baseline+ SCS.** Accuracy peaks at 63.1 for (r=0.2,m=6). When fixing r=0.2, performance first rises as the number of resampled trajectories grows (63.0 to 63.1 when truncation ratio from 0.2 to 0.4). then slips when the number of resampled trajectories larger (r=0.8,63.2).
>
> Nearly all three algorithms every (r,m) configuration still outperforms its vanilla counterpart by 0.6–2.2 pp, indicating that the consistency reward delivers a uniform benefit on different algorithms and hyperparameter settings.
>
> **W3: Lack of verification for assumptions.**
>
> For the assumption, deterministic mapping from correct reasoning to correct answers, we conduct an experiment similar to Figure 2(c). First, we manually select 100 cases which are solved with correct reasoning trajectories by Qwen2.5-VL-7B-Instruct for each benchmark. Then we remove the final option answer part (e.g., Answer: A.) for each initial response, and continue to generate from the truncation point. For each case, we do 4 resamples and count for the average number of final options for each question. The results are as follows:
>
> **Table R4. The average number of final options for each question in different benchmarks.**
> | **Benchmark**     | **M3CoT** | **MathVision** | **MMMU-Val** | **ScienceQA** | **MathVerse** | **WeMath** |
> |:-----------------:|:--------:|:--------------:|:------------:|:-------------:|:-------------:|:----------:|
> | **Nums**    |   1.0    |      1.1       |     1.1      |      1.0      |      1.0      |    1.0     |
>
> The results show that nearly all resamplings finish with the exact correct answer, illustrating that when the model follows the correct  trajectory, it almost certainly arrives at the correct option. We think this experiment can justify our assumption of the deterministic mapping from correct reasoning to correct answers.
>
> For the claimed differentiability of the consistency score, we have answered in W1, please refer to it.
>
> **W4: Comments Suggestions And Typos**
>
> Thank you for catching these issues. We will perform a thorough proof‑reading and correct all typos and formatting errors in the revised version.

---

> > ### Comment · Reviewer_bNXZ · 2025-08-04
> >
> > thanks for the detailed response, which indeed addresses my concerns.

---

### Official Review · Reviewer_QnZb · 2025-07-06

**Clarity:** 3
**Significance:** 3
**Originality:** 2
**Rating:** 4
**Confidence:** 4

**Summary:**

This paper addresses a prevalent flaw in outcome-reward reinforcement learning (RL) for multimodal large language models (MLLMs) in the multiple-choice setting: the tendency of models to receive full reward even when correct answers are obtained through reasoning that is unfaithful or inconsistent. The authors propose Self-Consistency Sampling (SCS), which augments outcome reward with a consistency-based signal computed by generating visual perturbations and truncated resamplings of the reasoning trajectory, assigning higher rewards to answers that are robust to these perturbations. Experimental results on six widely-used multimodal benchmarks demonstrate that incorporating SCS into existing outcome-reward RL algorithms results in notable accuracy improvements (up to 7.7 percentage points), with limited additional computational cost.

**Questions:**

1. The method is only demonstrated on datasets and tasks filtered for multiple-choice with multmodal input. While this is the target scenario, it reduces claims to a narrower setting, and limits immediate claims of generality for broader forms of reasoning, fill-in-the-blank, or free-form generation. Is there a way to generalize methodology？
2. How does SCS handle settings where the answer is open-ended or contains substantial ambiguity (e.g., free-form text rather than fixed choices)? Can the authors offer evidence or discuss expected performance and pitfalls in these domains?

**Ethical Concerns:**

["NO or VERY MINOR ethics concerns only"]

**Limitations:**

Yes

**Quality:**

3

**Strengths And Weaknesses:**

Strengths
1. The identification of unfaithful reasoning as a consequence of outcome-only rewards is well-motivated and empirically confirmed
2. SCS is conceptually simple, generic, and easily pluggable with a variety of outcome-reward RL algorithms
3. The paper offers extensive experiments over six benchmarks, covering a range of mathematical, scientific, and general reasoning tasks
4. Paper writing is clear and easy to read

Weaknesses
1. The work combines and adapts ideas of self-consistency, truncation-resampling, and visual perturbation in a manner specific to outcome-reward RL. While the integration is justified, substantial parts of the formulation (e.g., sampling, self-consistency reward, trajectory resampling) are inspired by prior literature (see references), and the paper could be clearer in distinguishing what is adopted, what is adapted, and what is new.
2. The consistency reward is based only on trajectory agreement, not external ground truth or human-like logic, so models might converge to internally consistent yet incorrect or non-humanlike reasoning.
3. While Figure 4 is informative for one example, a more systematic analysis (such as human annotation of faithfulness or error breakdown on a larger random sample) would strengthen claims about improved reasoning reliability.
4. Except for “statistical significance” being mentioned once (Section 4.2), scores in Table 2 and Table 3 are presented as point estimates without confidence intervals or statistical testing details.

---

> ### Author Rebuttal · Authors · 2025-07-30
>
> We sincerely thank you for your thoughtful feedback. We  appreciate the recognition of our work’s strengths, including the motivation, the simplicity and generality of our method, the adequacy of the experiments and the clarity of the writing. Besides, we have carefully replied your questions and concerns below.
>
> **W1: Separating Established Techniques from Our Novel Contributions**
>
> Thank you for the opportunity to clarify. Below we separate what we adopt, what we adapt, and what is new in our method:
>
> 1. Adopted ideas (directly reused)
>    - Self‑consistency sampling — drawing multiple reasoning trajectories and comparing their answers, as explored by Wang [1], Wang [2], and Xu[3].
>    - Perturbation for robustness — adding Gaussian noise to images to test semantic stability [4,5].
>
> 2. Adapted ideas (repurposed for RL)
>    - Truncation–resampling: originally a few‑shot prompting trick, we reinterpret it for RL by freezing the high‑confidence prefix of a trajectory and resampling only the tail, turning it into a low‑variance exploration scheme.
>    - Visual perturbation: instead of data augmentation, we use perturbations to create paired trajectories whose agreement becomes a training signal, not merely an evaluation metric.
>
> 3. New contribution (unique to this work)
>    - We introduce a self‑consistency reward: a differentiable signal that measures agreement across perturbed, truncation‑resampled trajectories and feeds this value directly into policy gradients. This shifts consistency from post‑hoc voting to an intrinsic learning objective, enabling the policy to shape its own trajectory distribution for greater robustness—an approach that, to our knowledge, has not appeared in outcome‑reward RL literature.
>
> We will revise the paper to make these three layers—**adopt**, **adapt**, **innovate**—explicit and add the missing citations noted above.
>
> **W2: Consistency rewarding may drive the model toward coherent but factually wrong reasoning**
>
> Thank you for raising this important point. As mentioned in our paper, we address this concern in two ways that together prevent the models drifting toward “consistent but wrong” solutions:
> 1. First, acc reward and consistency reward are both applied. Thus the accuracy component continually lead the policy to externally correct behavior, while the consistency term  lead toward reliables strategies instead of replacing the ground truth.
> 2. Second, we apply a small coefficient c=0.4 for consistency reward, ensuring that the model cannot “hack” its way to high reward by neglecting real task performance.
>
> **W3: The systematic analysis of about improved reasoning reliability with SCS.**
>
> Thanks for pointing out this problem. To give concrete proof that our SCS method improves reasoning reliability, we conducted a quantitative analysis. To be specific, for each benchmark we randomly sampled 100 cases that  models answered with the correct option before and after training with SCS. Then we manually checked each case whether the model's output is aligned with publicly provided solutions and count times of unfaithful reasoning. Besides, we also asked two strong closed‑source LLMs (OpenAI o3‑mini and Gemini 2.5 Flash) to rate the same traces. The results are shown below:
>
> **Table R1. Quantitative analysis of the improvement in reasoning reliability after applying SCS.** The numbers in the table represent the occurrences of unfaithful reasoning in 100 correctly answered questions.
>
> | **Judger**        | **Model**             | **Avg**         | **M3CoT** | **MathVision** | **MMMU-Val** | **ScienceQA** | **MathVerse** | **WeMath** |
> |:-----------------:|:----------------------|:---------------:|:--------:|:--------------:|:------------:|:-------------:|:-------------:|:----------:|
> | Human             | Qwen2.5-VL-7B         | 25.0            |   14     |      64        |     30       |      11       |      16       |    15      |
> |                   | Qwen2.5-VL-7B-SCS     | 21.2 (−15.2%)   |   12     |      56        |     25       |       9       |      12       |    13      |
> | o3-mini           | Qwen2.5-VL-7B         | 22.0            |   11     |      55        |     35       |       8       |      12       |    11      |
> |                   | Qwen2.5-VL-7B-SCS     | 19.0 (−13.6%)   |    9     |      49        |     29       |       7       |      10       |    10      |
> | Gemini 2.5 Flash  | Qwen2.5-VL-7B         | 23.0            |   12     |      57        |     34       |       8       |      13       |    14      |
> |                   | Qwen2.5-VL-7B-SCS     | 19.7 (−14.3%)   |   10     |      50        |     29       |       8       |      10       |    11      |
>
> The results show a around 15% in faithfulness across all three datasets, demonstrating that our central claim that SCS not only boosts answer accuracy but also produces more reliable reasoning. We will include this table in the revised version of our paper.
>
> **W4: Lack of Confidence Intervals in Reported Scores**
>
> Thank you for pointing this out—we now include full statistical reporting based on our repeated‑run data.
> Based on Tables 2 and 3, we carried out three additional runs under the same experimental setup as in the paper; the resulting scores and their 95% confidence intervals are presented in the table below.
>
> **Table R2. 95 % confidence intervals for the results of different RL algorithm experiments.**
> | **Algorithm**            | **Run 1** | **Run 2** | **Run 3** | **Mean** | **95% CI (n = 3)** |
> |:------------------------:|:--------:|:--------:|:--------:|:--------:|:------------------:|
> | GRPO                     |  64.5    |  64.4    |  64.6    |  64.5    |   64.5 ± 0.3       |
> | REINFORCE++-baseline     |  63.0    |  63.1    |  62.8    |  62.6    |   62.9 ± 0.6       |
> | REINFORCE++              |  62.9    |  63.4    |  63.0    |  63.1    |   63.0 ± 0.4       |
> | RLOO                     |  65.5    |  65.1    |  64.7    |  65.1    |   65.1 ± 1.0       |
>
> **Table R3. 95 % confidence intervals for the results of ablations experiments for two components in SCS (TR = Truncation–Resampling, VP = Visual-Perturbation).**
>
> | **TR** | **VP** | **Run1** | **Run2** | **Run3** | **Mean** | **95% CI (n = 3)** |
> |:------:|:------:|:--------:|:--------:|:--------:|:--------:|:------------------:|
> | ✗      | ✗      |  57.8    |  57.8    |  57.7    |  57.8    |    57.8 ± 0.1      |
> | ✓      | ✗      |  63.0    |  62.7    |  63.0    |  62.9    |    62.9 ± 0.6      |
> | ✗      | ✓      |  62.8    |  63.4    |  62.6    |  63.1    |    62.9 ± 1.0      |
> | ✓      | ✓      |  65.5    |  65.1    |  64.7    |  65.1    |    65.1 ± 1.0      |
>
> As the table shows, even with only three repeated runs, the confidence intervals remain small, indicating a robust positive effect. We will conduct more runs and update the table in our next version.
>
> **Q1: Settings confined to multimodal multiple‑choice tasks limits evidenced generality**
>
> Thank you for your thoughtful question. Our goal is to curb unfaithful reasoning.  In practice, we observe this failure mode most frequently in multiple‑choice settings, where the models can exploit option‑specific answers.  In more open‑ended formats—fill‑in‑the‑blank or free‑form generation—the space of outputs is much larger and such shortcut patterns appear far less often, so the original RLVR algorithm alone already behaves robustly [3,7,8].  And In Figure 2(a) we have conducted an experiment to verify this. For that reason, we concentrated our methodological contribution  on multiple‑choice tasks, where it delivers the most benefit.
>
> At the same time, we think that the multi-choice paradigm is not that narrow: it takes the a large part of  RL datasets such as M3cot, ScienceQA, comt and so on. These cover natural images, diagrams, videos, and so on; domains range from science to daily-life questions. Therefore, although our experiments are focused on multi-choice questions, the application scope spans a wide variety of real‑world reasoning scenarios.
>
> **Q2: Limited evidence on how SCS candles open‑ended scenarios.**
>
> We greatly appreciate your feedback. As mentioned above, in open‑ended settings such as fill‑in‑the‑blank or free‑form generation, the output space is so large that brittle shortcut patterns become exceedingly rare. Consequently, the base RLVR algorithm already exhibits strong robustness  [3,6]. While our SCS reward can still be applied in principle, we expect its incremental benefit to be smaller because agreement among independently sampled long‑form answers is naturally harder to achieve and hard to be verified. Extending SCS to these domains is an interesting direction we leave for future work.
>
>
>
> [1] Wang, X., Wei, J., Schuurmans, D., Le, Q., Chi, E., Narang, S., ... & Zhou, D. (2022). Self-consistency improves chain of thought reasoning in language models. arXiv preprint arXiv:2203.11171.
>
> [2] Wang, H., Prasad, A., Stengel-Eskin, E., & Bansal, M. (2024). Soft self-consistency improves language model agents. arXiv preprint arXiv:2402.13212.
>
> [3] Wang, X., Wei, J., Schuurmans, D., Le, Q., Chi, E., Narang, S., ... & Zhou, D. (2022). Self-consistency improves chain of thought reasoning in language models. arXiv preprint arXiv:2203.11171.
>
> [4] Cohen, J., Rosenfeld, E., & Kolter, Z. (2019, May). Certified adversarial robustness via randomized smoothing. In international conference on machine learning (pp. 1310-1320). PMLR.
>
> [5] Hendrycks, D., & Dietterich, T. (2019). Benchmarking neural network robustness to common corruptions and perturbations. arXiv preprint arXiv:1903.12261.
>
> [6] Yue, Y., Chen, Z., Lu, R., Zhao, A., Wang, Z., Song, S., & Huang, G. (2025). Does reinforcement learning really incentivize reasoning capacity in llms beyond the base model?. arXiv preprint arXiv:2504.13837.

---

> ### Author Response · Authors · 2025-08-07
>
> Thank you again for your valuable feedback. We have responded to your questions in the rebuttal. We hope our response addresses your concerns. If you have any further questions or require clarification, we would be happy to provide additional information.

---

### Official Review · Reviewer_xohU · 2025-07-07

**Clarity:** 3
**Significance:** 2
**Originality:** 3
**Rating:** 4
**Confidence:** 3

**Summary:**

This paper proposes Self-Consistency Sampling (SCS), a method that calculates the outcome reward by taking consistency across multiple rollouts into account for improving RL fine-tuning of MLLMs. This paper illustrates unfaithful reasoning phenomenon where the answer is correct but the reasoning process is wrong. And, this paper shows that outcome reward-based RL fine-tuning cannot properly reflect these cases. More specifically, SCS calculates the consistency-based reward by introducing visual perturbation and re-generating truncated trajectories. This paper evaluates SCS on six benchmarks including M^3CoT, ScienceQA, MathVision, and We-Math, MMMU, and MathVerse. The experiment results show that SCS applied to RLOO achieves performance improvements by about 7.7%.

**Questions:**

- Q1. What percentage of cases are there where the answer is correct but the reasoning process is wrong?

- Q2. According to Table 2, SCS performs well, when being applied to a specific RL method such as RLOO. What do the authors think is the reason for this?

- Q3. According to Table 3 (i.e., Ablation studies of component effectiveness), both truncation-resampling (from 57.8% to 63.0%) and visual-perturbation (from 57.8% to 62.8%) are effective in performance improvements. Do the authors think that visual-perturbation is necessary for SCS? Then, why do the authors think the visual-perturbation is necessary?

**Ethical Concerns:**

["NO or VERY MINOR ethics concerns only"]

**Final Justification:**

I maintain my initial rating: 4: Borderline accept. This paper proposes Self-Consistency Sampling (SCS), a method that calculate the outcome reward by leveraging consistency over multiple rollouts for RL fine-tuning MLLMs. In my initial review, I pointed out three weak points and raised three questions: W1. generality of SCS across diverse RL algorithms, W2. principles of algorithm design, W3. additional computation cost, Q1. ratio of unfaithful reasoning phenomenon, Q2. reasons for the generality of SCS, and Q3. reasons for the necessity of visual-perturbation. The authors provided thoughtful responses to my comments and questions. The responses helped me understand this paper better. This paper empirically shows that SCS can alleviate the unfaithful reasoning phenomenon, eventually increasing the accuracy of reasoning tasks. However, this paper has some limitations that SCS provides performance improvements on some RL algorithms with large variance such as RLOO.

**Limitations:**

The authors have provided the limitations of their work in the second paragraph (i.e., Limitations) of Section 5 (i.e., Conclusion). However, the paragraph consists of only one sentence. It would be better to discuss the limitation in more detail.

**Paper Formatting Concerns:**

There don’t seem to be any major formatting issues in this paper.

**Quality:**

2

**Strengths And Weaknesses:**

The strengths of this paper can be summarized as follows:
- S1. This paper is well motivated by illustrating unfaithful reasoning phenomenon.
- S2. It is interesting to estimate the consistency-based reward by generating multiple trajectories.

The weaknesses of this paper can be summarized as follows:
- W1. According to Table 2, the proposed method Self-Consistency Sampling (SCS) performs well when only being applied to a specific RL method. More specifically, the SCS applied to RLOO increases overall accuracy by 7.7% (i.e., RLOO: 57.8%, RLOO + SCS: 65.5%). However, the SCS applied to other RL methods such as GRPO and REINFORCE++ provide very limited performance improvements (i.e., GRPO: 63.6%, GRPO + SCS: 64.5%).

- W2. The design of SCS appears to be based on experimentation rather than principles. What is the advantage of truncation-resampling over total-resampling? Also, is visual-perturbing necessary for SCS? Is visual-perturbing for robust reasoning process?

- W3. Even though SCS can improve the performance, it incurs additional computation costs.

---

> ### Author Rebuttal · Authors · 2025-07-30
>
> Thank you for the feedback and your appreciation for our motivation and our Self-Consistency Sampling method. Additionally, you raised several important concerns and questions, which we address in detail below.
>
> **Q1: The percentage of  unfaithful reasoning phenomenon.**
>
> Thanks for your question. For the same benchmarks we tested in the paper, we manually check 100 items which are correctly answered by Qwen2.5-VL-7B and count the number of times this situation occurs. We have illustrated the results in Appendix, and we put it again below:
>
> **Table R1. Statistics of unfaithful reasoning across different benchmarks.** The table show the occurrences of unfaithful reasoning in 100 correctly answered questions.
>
> | **Benchmark**              | **M3CoT** | **Mathvision** | **MMMU-Val** | **ScienceQA** | **MathVerse** | **WeMath** |
> |:--------------------------:|:--------:|:--------------:|:-------:|:-------------:|:-------------:|:----------:|
> | **Nums** |   14     |      64        |   30    |      11       |      16       |    15      |
>
> Comprehensively analyzing the table and Figure 2c,  We can conclude that the unfaithfull reasoning phenomenon is universal, and it is notable in some benchmarks (64% unfaithfull reasoning rate for mathvision), which further confirms our motivation.
>
> **Q2 & W1: The performance gains from SCS are not consistent across different RL algorithms.**
>
> Thanks for your very insightful question. The effectiveness of SCS appears to be limited to a specific algorithm, and one of the reasons is that SCS works by introducing resampling and perturbation which reduce the variance of gradient estimation. RLOO, which uses a leave-one-out advantage, tends to suffer from higher variance, so the variance reduction brought by SCS yields more significant gains. In contrast, methods like GRPO and REINFORCE++ already incorporate normalization that reduces variance, leaving limited room for further improvement by SCS.
> To be specific, we calculate the variance of advantage. Let the episodic rewards  $R_1, \ldots, R_N$ with mean $\mu$ and variance $\sigma_R^2 $. For RLOO, the advantage is
> $$A_{\mathrm{RLOO}}^{(i)}=R^{(i)}-\frac{1}{N-1} \sum_{j \neq i} R^{(j)}$$
> Because $R_i$ is not included in the leave‑one‑out baseline, the two terms are independent. Hence,
> $$\operatorname{Var}\left[A_{\mathrm{RLOO}}\right]=\sigma_R^2\left(1+\frac{1}{N-1}\right)$$
> For GRPO and REINFORCE++, since they have already applied normalization, the variance is small. Because RLOO inherently has high variance, the variance‑reduction effect of SCS is significant, yielding a clearer improvement in model performance. In contrast, the other two algorithms already exhibit low variance, so the benefit is much less noticeable.
>
> **W2: The advantage of truncation-resampling over total-resampling.**
>
> Thank you for pointing out this important detail. Truncation‑resampling strikes a compute‑efficient balance: instead of regenerating full trajectories like total‑resampling, we  resample only the uncertain tail, cutting sampling time and GPU memory. What' more, this efficiency does not come at the expense of quality—our ablation in Figure 5 shows that when we change the truncation ratio r, the best performance occurred at r=0.8, higher than r=0.2 (nearly total‑resampling).  In a word, our SCS both saves resources and improves performance.
>
> **Q3& W2: Explanations of the principles and necessity of visual‑perturbation (VP).**
>
> Thank you for your thoughtful question. For the explanation of visual‑perturbation, we think it can be seen as a part of resampling. These perturbed inputs further de‑correlate the individual rollouts, so the results can aggregate more independent samples rather than near‑duplicates.
> We still think VP is necessary and effective. First, Table 3 already confirms that both truncation‑resampling (TR) and visual‑perturbation (VP) are individually beneficial —raising accuracy from 57.8 % to 63.0 % and 62.8 %, and their combination yields an additional boost beyond either component alone, reaching 65.5 %. Furthermore, our case studiy shows that after applying the visual‑perturbation method, the model not only retains its understanding of the image’s semantics but also gains a stronger capacity for fine‑grained reasoning. So using them together can bring further performance gains.
>
> **W3: Additional computation costs caused by SCS.**
>
> Thank you for bringing up this issue. The extra overhead introduced by SCS method lies almost entirely in the sampling phase Leveraging modern high‑performance inference engines such as vLLM, these process are batched and run in parallel, so the wall‑clock impact grows sub‑linearly with N and remains well‑controlled.
> Under identical hyperparameters on 8 * A100 GPU (N = 4, truncation ratio=0.8), we observed:
>
> **Table R2. Comparison of training time costs after applying SCS.**
>
> | Configuration     | Training Time | Time Change   | Scores |  Improvement |
> |:-----------------:|:-------------------:|:----------------------:|:------:|:------------------------:|
> | Baseline          | 12.5 h              | —                      | 57.8   | —                        |
> | Baseline + SCS    | 17.2 h              | ≈ +38 %                | 65.5   | +7.7                     |
>
> Thus, with advanced inference backends, the time cost of SCS is both predictable and acceptable given the performance gains it enables.
>
> **Limitations:**
>
> As for limitations, our SCS has not been extensively applied to LLMs and more MLLMs to verify its generality. The SCS method is primarily suited for multiple-choice style training datasets and does not adapt well to open-ended or fill-in-the-blank questions. Therefore, it is currently mainly applicable to outcome-based RL algorithms. Additionally, the method should be tested on more datasets and benchmarks.

---

> ### Comment · Reviewer_xohU · 2025-08-07
>
> Thank you for providing thoughtful responses to my questions and comments. The responses helped me understand the paper better. Here are my thoughts on the author responses:
> - It is interesting that there are 64 unfaithful reasoning processes in 100 correctly answers questions in MathVision dataset. This observation may have led the method of visual-perturbation.
> - It is unfortunate that Self-Consistency Sampling (SCS) is effective only for RL algorithms with high variance such as RLOO.
> - Both truncation-resampling and visual-perturbation appear to be effective. However, it is unfortunate that the effect is not amplified when used together.
>
> After reviewing the author responses, I maintain my initial score.

---

### Note · Authors · 2025-08-12

We sincerely thank the reviewers for their valuable feedback and are encouraged by their recognition of our work’s strengths as follows:
1. **Clear problem motivation and identification** – The paper is well-motivated by clearly illustrating the phenomenon of unfaithful reasoning in outcome-based RL.
2. **Novel yet simple solution** – The proposed Self-Consistency Sampling (SCS) is simple, generic, and easily pluggable into various outcome-reward RL algorithms.
3. **Strong experimental validation** – Extensive experiments across six benchmarks and four RL algorithms, covering mathematical, scientific, and general reasoning tasks, consistently demonstrate performance gains.
4. **Comprehensive analysis** – Detailed ablation studies show the independent and combined effects of truncation–resampling and visual perturbation, as well as robustness under hyperparameter sweeps.

At the same time, we concluded the common concerns raised by several reviewers. In our rebuttal and the updated manuscript, we have addressed these points as follows:
1. **Quantitative analysis of improved reasoning reliability with SCS** – We conducted a quantitative analysis using both human evaluation and two advanced MLLMs (o3-mini and Gemini 2.5 Flash). The results show that the phenomenon of unfaithful reasoning is common, and our method can reduce its occurrence by approximately 15%.
2. **Validation of SCS on models with different architectures and scales** – We ran the protocol on two additional models with different architectures and scales (Qwen2.5-VL-3B-Instruct and InternVL3-8B). The results confirm that SCS is applicable to a broader range of model architectures and parameter sizes.
3. **Inclusion of confidence intervals in reported scores** – We have added full statistical reporting based on repeated runs, which indicates a robust positive effect.
4. **Analysis of the effect of visual perturbation** – We provided a detailed explanation of the visual perturbation component and demonstrated its effectiveness through experimental results.
5. **Refined the theoretical analysis** – We clarified the assumptions, restated the modeling method and refined the use of notations.

We believe these clarifications and improvements have significantly strengthened the paper. We again appreciate the reviewers’ insightful comments and constructive guidance.

---

### Decision · Program_Chairs · 2025-09-17

**Decision:**

Accept (poster)

**Comment:**

(a) Summary of Scientific Claims and Findings

This paper addresses a key limitation in outcome-reward-based reinforcement learning (RL) for multimodal large language models (MLLMs): the issue of "unfaithful reasoning," where a model produces a faulty reasoning trajectory but still arrives at the correct answer, thereby receiving an undeservedly high reward. The authors propose Self-Consistency Sampling (SCS), a method designed to improve reasoning faithfulness. SCS works by generating multiple reasoning trajectories from a common prefix. These trajectories are diversified through two mechanisms: (i) truncation-and-resampling of the reasoning chain and (ii) adding small visual perturbations to the input image. The agreement (or consistency) among the final answers from these trajectories is quantified into a differentiable consistency score, which is then incorporated as an additional reward signal during RL training. The primary claim is that by rewarding consistency, SCS encourages the model to learn more robust and faithful reasoning paths. The authors demonstrate empirically that integrating SCS into four different RL algorithms (RLOO, GRPO, REINFORCE++) leads to significant accuracy improvements (up to 7.7 percentage points) across six multimodal reasoning benchmarks.

(b) Strengths of the Paper

Based on the reviews, the paper's main strengths are:

Clear Problem Motivation: All reviewers acknowledged that the paper clearly identifies and motivates an important and often-overlooked problem in outcome-based RL for MLLMs—the unfaithful reasoning phenomenon.

Simple and General Solution: The proposed SCS method is conceptually simple, intuitive, and designed as a "plug-and-play" module that can be easily integrated into various existing outcome-reward RL frameworks.

Strong Empirical Validation: The authors conducted extensive experiments across six benchmarks and four RL algorithms, demonstrating consistent performance gains. The ablation studies effectively disentangle the contributions of the two main components of SCS (truncation-resampling and visual perturbation).

Thorough Rebuttal and Analysis: The authors provided a comprehensive rebuttal with substantial new experiments that addressed most of the reviewers' initial concerns. This included quantitative analysis of reasoning faithfulness, experiments on additional model architectures and scales, and the addition of confidence intervals for statistical robustness.

(c) Weaknesses of the Paper

The initial submissions had several weaknesses that were largely, though not entirely, addressed during the rebuttal:

Lack of Quantitative Faithfulness Analysis: The central claim of improving reasoning reliability was initially supported only by qualitative examples. This was a major concern for reviewers uohx, QnZb, and pWKS.

Limited Generality and Statistical Rigor: Concerns were raised about the use of a single MLLM, the absence of confidence intervals in the results (uohx, QnZb), and the observation that performance gains were much more pronounced for the high-variance RLOO algorithm compared to others (xohU).

Clarity and Theoretical Soundness: Some reviewers (bNXZ, pWKS) found issues with clarity, including inconsistent notation, a misleading claim about a "differentiable" score, and theoretical assumptions that were not empirically validated in the original manuscript.

Remaining Concerns: Even after the rebuttal, Reviewer pWKS expressed reservations about the strength of the theoretical assumptions, suggesting they were not fully validated by the new experiments. Reviewer xohU noted that the method's effectiveness being tied to high-variance RL algorithms is a limitation.

(d) Most Important Reasons for Recommendation

I recommend acceptance for this paper. The primary reason is that the authors identify a significant problem and propose a practical, effective solution. The empirical results are strong and were made more robust during the rebuttal phase.

The authors' response to the reviews was very thorough. They went beyond simple clarifications and conducted multiple new, non-trivial experiments to provide quantitative evidence for their core claims. They added human and LLM-based evaluations to demonstrate improved reasoning faithfulness, tested their method on two additional models to show generality, and provided statistical analysis with confidence intervals. This proactive engagement significantly strengthened the paper and addressed the most critical weaknesses identified by the reviewers. While minor concerns about the theoretical assumptions and variable effectiveness across RL algorithms remain, the paper's practical contribution, strong empirical backing, and clear motivation make it a valuable addition to the field.

(e) Summary of Discussion and Rebuttal

The rebuttal period was productive and pivotal for this paper.

Points raised by reviewers:

Reviewers uohx, QnZb, and pWKS requested quantitative evidence that SCS improves reasoning faithfulness, not just final-answer accuracy.

Reviewers uohx and QnZb pointed out the lack of confidence intervals or multiple runs to establish statistical significance.

Reviewer uohx asked about generality to other MLLMs.

Reviewer xohU questioned why the performance gains were much larger for RLOO than for other RL algorithms.

Reviewers bNXZ and pWKS raised issues with theoretical clarity, including the "differentiable score" claim and the validation of core assumptions.

Reviewers xohU and uohx asked for an analysis of computational overhead.

Authors' responses:

Faithfulness: The authors conducted a new analysis using human annotators and two strong MLLMs (o3-mini, Gemini 2.5 Flash), showing that SCS reduced the rate of unfaithful reasoning by approximately 15%. This directly addressed a major concern.

Statistical Rigor: They provided new tables with results from three independent runs, including means and 95% confidence intervals, confirming the robustness of their findings.

Generality: They ran experiments on two additional models (Qwen2.5-VL-3B-Instruct and InternVL3-8B), both showing performance gains with SCS.

Varying Gains: They offered a plausible explanation that SCS's variance-reduction effect is more impactful on high-variance algorithms like RLOO, whereas algorithms like GRPO already have variance-control mechanisms.

Theoretical Clarity: They clarified the "differentiable" terminology, promised to fix all notational inconsistencies, and provided a new experiment to empirically support their assumption that correct reasoning trajectories reliably lead to correct answers.

Overhead: They presented a clear comparison showing SCS added ~38% to training time, which they argued was a reasonable trade-off for the ~7.7% accuracy gain.

The addition of quantitative analysis on faithfulness and statistical significance was crucial and satisfied reviewers uohx and QnZb, with Reviewer uohx increasing their score. The new experiments on model generality also strengthened the paper in my opinion. While Reviewer pWKS remained skeptical about the theoretical assumptions, they acknowledged the new evidence was interesting and raised their score, but I think the method is still useful even if some of the theoretical assumptions don't apply in some practical settings. The explanation for the performance variation across RL algorithms was accepted as reasonable by Reviewer xohU, which seems sensible to me as well. Overall, the authors successfully addressed the most critical weaknesses, transforming borderline scores into a clearer case for acceptance. The paper's strengths outweigh its weaknesses in my opinion so I recommend acceptance.